# Instance-Conditioned GAN Data Augmentation for Representation Learning

## Abstract

Data augmentation has become a crucial component to train state-of-the-art visual representation models. However, handcrafting combinations of transformations that lead to improved performances is a laborious task, which can result in visually unrealistic samples. To overcome these limitations, recent works have explored the use of generative models as learnable data augmentation tools, showing promising results in narrow application domains, e.g., few-shot learning and low-data medical imaging. In this paper, we introduce a data augmentation module, called $DA_{IC\text{-}GAN}$, which leverages instance-conditioned GAN generations and can be used off-the-shelf in conjunction with most state-of-the-art training recipes. We showcase the benefits of $DA_{IC\text{-}GAN}$ by plugging it out-of-the-box into the supervised training of ResNets and DeiT models on the ImageNet dataset, and achieving accuracy boosts up to between 1%p and 2%p with the highest capacity models. Moreover, the learnt representations are shown to be more robust than the baselines when transferred to a handful of out-of-distribution datasets, and exhibit increased invariance to variations of instance and viewpoints. We additionally couple $DA_{IC\text{-}GAN}$ with a self-supervised training recipe and show that we can also achieve an improvement of 1%p in accuracy in some settings. We open-source the code at `anonymous.url` to encourage reproducibility and further future explorations. With this work, we strengthen the evidence on the potential of learnable data augmentations to improve visual representation learning, paving the road towards non-handcrafted augmentations in model training.

## 1 Introduction

Recently, deep learning models have been shown to achieve astonishing results across a plethora of computer vision tasks when trained on *very large* datasets of hundreds of millions datapoints (Alayrac et al., 2022; Gafni et al., 2022; Goyal et al., 2022; Radford et al., 2021; Ramesh et al., 2022; Saharia et al., 2022; Zhang et al., 2022). Oftentimes, however, large datasets are not available, limiting the performance of deep learning models. To overcome this limitation, researchers explored ways of artificially increasing the size of the training data by transforming the input images via *handcrafted* data augmentations. These augmentation techniques consist of heuristics involving different types of image distortion (Shorten & Khoshgoftaar, 2019), including random erasing (DeVries & Taylor, 2017b; Zhong et al., 2020), and image mixing (Yun et al., 2019; Zhang et al., 2017). It is important to note that all current state-of-the-art representation learning models seem to benefit from such complex data augmentation recipes as they help regularizing models – e.g., vision transformers trained with supervision (Dosovitskiy et al., 2021; Touvron et al., 2021; Steiner et al., 2021; Touvron et al., 2022) and models trained with self-supervision (Chen et al., 2020a; He et al., 2020; Caron et al., 2020; 2021; Grill et al., 2020). However, coming up with data augmentation recipes is laborious and the augmented images, despite being helpful, often look unrealistic, see first five images in Figure 1. Such a lack of realism is a sub-optimal effect of these heuristic data augmentation strategies, which turns out to be even detrimental when larger training datasets are available (Steiner et al., 2021) and no strong regularization is needed.

To this end, researchers have tried to move away from training exclusively on real dataset samples and their corresponding hand-crafted augmentations, and have instead explored increasing the dataset sizes with generative model samples (Frid-Adar et al., 2018; Bowles et al., 2018; Ravuri & Vinyals, 2019; Zhang et al.,

| Original sample | RandomCrop | RandAugment | MixUp | CutMix | IC-GAN |
|---|---|---|---|---|---|

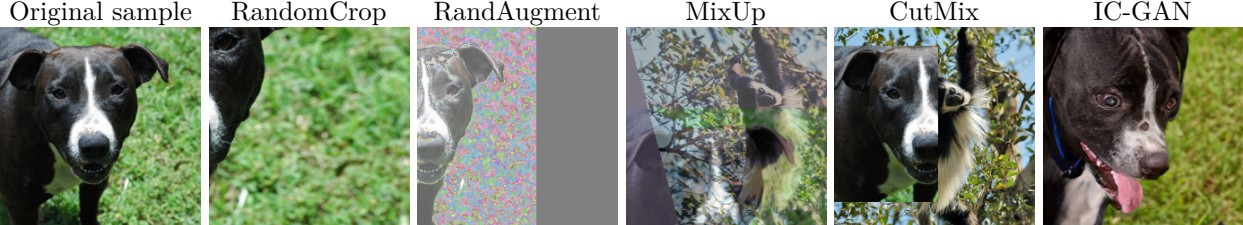

Figure 1: Visual comparison of various hand-crafted data augmentations and one IC-GAN generation, using a sample from ImageNet as input.

2021; Li et al., 2021). A generative model can potentially provide infinitely many synthetic image samples; however, the quality and diversity of the generated samples is usually a limiting factor, resulting in moderate gains for specific tasks like image segmentation (Zhang et al., 2021; Li et al., 2021) and in poor performance in standard image classification benchmarks (Ravuri & Vinyals, 2019). With the advent of photorealistic image samples obtained with generative adversarial networks (GAN) (Goodfellow et al., 2014), researchers have explored the use of GAN-based data augmentation (Antoniou et al., 2018; Tritrong et al., 2021; Mao et al., 2021; Wang et al., 2021). However, none of these approaches has shown improvement when applied to large-scale datasets, such as ImageNet (Deng et al., 2009), in most cases due to a lossy and computationally intensive GAN-inversion step.

In this paper, we study the use of Instance-Conditioned GAN (IC-GAN) (Casanova et al., 2021), a generative model that, conditioned on an image, generates samples that are semantically similar to the conditioning image. Thus, we propose to leverage IC-GAN to generate plausible augmentations of each available data-point and design a module, called $DA_{IC\text{-}GAN}$, that can be coupled off-the-shelf with most supervised and self-supervised data augmentation strategies and training procedures. We validate the proposed approach by training supervised image classification models of increasing capacity on the ImageNet dataset and evaluating them in distribution and out-of-distribution. Our results highlight the benefits of leveraging $DA_{IC\text{-}GAN}$, by outperforming strong baselines when considering high-capacity models, and by achieving robust representations exhibiting increased invariance to viewpoint and instance. We further couple $DA_{IC\text{-}GAN}$ with a self-supervised learning model and show that we can also boost its performance in some settings.

Overall, the contributions of this work can be summarized as follows:

- We introduce $DA_{IC\text{-}GAN}$, a data augmentation module that combines IC-GAN with handcrafted data augmentation techniques and that can be plugged off-the-shelf into most supervised and self-supervised training procedures.

- We find that using $DA_{IC\text{-}GAN}$ in the supervised training scenario is beneficial for high-capacity networks, e.g., ResNet-152, ResNet-50W2, and DeIT-B, boosting in-distribution performance and robustness to out-of-distribution when combined with traditional data augmentations like random crops and RandAugment.

- We extensively explore $DA_{IC\text{-}GAN}$ 's impact on the learned representations, we discover an interesting correlation between per-class FID and classification accuracy, and report promising results in the self-supervised training of SwAV when not used in combination with multi-crop.

- We release the code-base and trained models at `anonymous.url` to foster further research on the usage of IC-GAN as a data augmentation technique.

## 2 Related Work

**Image distortion.** Over the past decades, the research community has explored a plethora of simple hand-designed image distortions such as zoom, reflection, rotation, shear, color jittering, solarization, and blurring — see Shorten & Khoshgoftaar (2019) and Perez & Wang (2017) for an extensive survey. Although

all these distortions induce the model to be robust to small perturbations of the input, they might lead to unrealistic images and provide only limited image augmentations. To design more powerful image distortions, the research community has started to combine multiple simple image distortions into a more powerful data augmentation schemes such as Neural Augmentation (Perez & Wang, 2017), SmartAugment (Lemley et al., 2017), AutoAugment (Cubuk et al., 2019), and RandAugment (Cubuk et al., 2020). Although these augmentation schemes oftentimes significantly improve model performance, the resulting distortions are limited by the initial set of simple distortions. Moreover, finding a good combination of simple image distortions is computationally intense since it requires numerous network trainings.

**Image mixing.** An alternative way to increase the diversity of augmented images is to consider multiple images and their labels. For example, CutMix (Yun et al., 2019) creates collages of pairs of images while MixUp (Zhang et al., 2017) interpolates them pixel-wise. In both cases, the mixing factor is regulated by a hyper-parameter, which is also used for label interpolation. However, these augmentation techniques directly target the improvement of class boundaries, at the cost of producing unrealistic images. We argue that unrealistic augmentations are a sub-optimal heuristic currently adopted as a strong regularizer, which is no longer needed when larger datasets are available, as shown in Steiner et al. (2021).

**Data augmentation with autoencoders.** To improve the realism of augmented images, some researchers have explored applying the image augmentations in the latent space of an autoencoder (AE). DeVries & Taylor (2017a) and Liu et al. (2018) proposed to interpolate/extrapolate neighborhoods in latent space to generate new images. Alternatively, Schwartz et al. (2018) introduced a novel way of training AE to synthesize images from a handful of samples and use them as augmentations to enhance few-shot learning. Finally, Pesteie et al. (2019) used a variational AE trained to synthesize clinical images for data augmentation purposes. However, most of above-mentioned approaches are limited by the quality of the reconstructed images which are oftentimes blurry.

**Data augmentation with generative models.** To improve the visual quality of augmented images, the community has studied the use of generative models in the context of both data augmentation and dataset augmentation. Tritrong et al. (2021); Mao et al. (2021) explored the use of instance-specific augmentations obtained via GAN inversion (Xia et al., 2022; Huh et al., 2020; Zhu et al., 2016), which map original images into latent vectors that can be subsequently transformed to generate augmented images (Jahanian et al., 2020; Härkönen et al., 2020). However, GAN inversion is a computationally intense operation and latent space transformations are difficult to control (Wang et al., 2019; 2021). Antoniou et al. (2018) proposed a specific GAN model to generate a realistic image starting from an original image combined with a noise vector. However, this work was only validated on low-shot benchmarks. Researchers have also explored learning representations using samples from a trained generative model exclusively (Shrivastava et al., 2017; Zhang et al., 2021; Li et al., 2022; Besnier et al., 2020; Li et al., 2021; Zhao & Bilen, 2022; Jahanian et al., 2022) as well as combining real dataset images with samples from a pre-trained generative model (Frid-Adar et al., 2018; Bowles et al., 2018; Ravuri & Vinyals, 2019), with the drawback of drastically shifting the training distribution. Finally, the use of unpaired image-to-image translation methods to augment small datasets was explored in Sandfort et al. (2019); Huang et al. (2018); Gao et al. (2018); Choi et al. (2019). However, such approaches are designed to translate source images into target images and thus are limited by the source and target image distributions.

**Data augmentation with latent neighbor images.** Another promising data augmentation technique uses neighbor images to create semantically-similar image pairs that can be exploited for multi-view representation learning typical of SSL. This technique was promoted in NNCLR (Dwibedi et al., 2021), an extension of the SSL model SimCLR (Chen et al., 2020a) to use neighbors, with some limitations due to the restricted and dynamic subset used for neighbor retrieval. Alternatively, Jahanian et al. (2022) explored the generation of neighbor pairs by using latent space transformations in conjunction with a pre-trained generative model. However, this model only uses generated samples to learn the representations and reports poor performance on a simplified ImageNet setup (training on $128 \times 128$ resolution images for only 20 epochs).

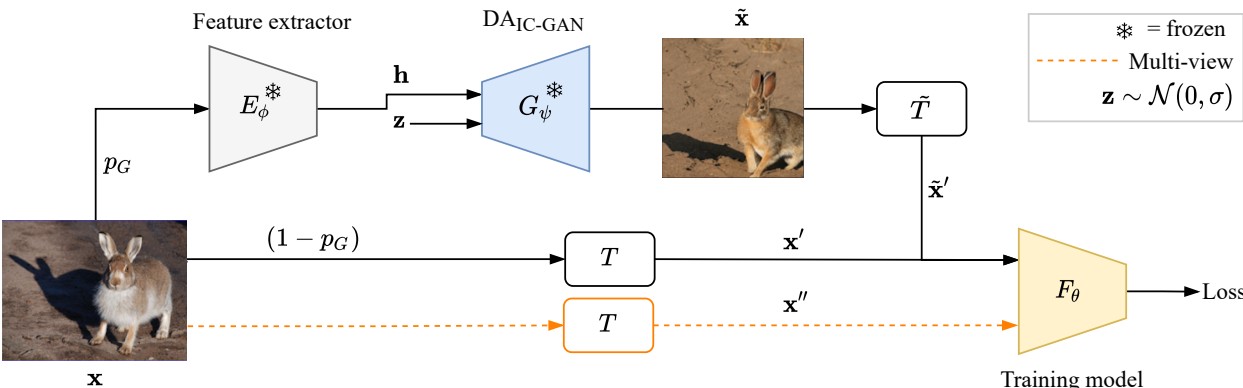

Figure 2: $\text{DA}_{\text{IC-GAN}}$ integration scheme to train a model $F_\theta$. For each image $\mathbf{x} \in \mathcal{D}$ we apply $\text{DA}_{\text{IC-GAN}}$, with probability $p_G$. When an image is IC-GAN-augmented, the representation $\mathbf{h}$ of the image is used as input to the generator, together with the Gaussian noise $\mathbf{z}$. The generated image $\tilde{\mathbf{x}}$ undergoes an additional sequence of handcrafted data augmentations, $\tilde{T}$. When $\text{DA}_{\text{IC-GAN}}$ is not applied, the standard handcrafted data augmentation, $T$, is applied to $\mathbf{x}$ to produce $\mathbf{x}'$. In the case of multi-view SSL training (orange branch), an additional view, $\mathbf{x}''$, is obtained by independently applying $T$ to the original image.

## 3  Methodology

### 3.1  Review of Instance-Conditioned GAN (IC-GAN)

Instance-conditioned GAN (IC-GAN) (Casanova et al., 2021) is a conditional generative model that synthesizes high quality and diverse images that resemble an input image used to condition the model. The key idea of IC-GAN is to model the data distribution as a mixture of overlapping and fine-grained data clusters, defined by each datapoint – or "instance" – and the set of its nearest neighbors. Training IC-GAN requires access to a dataset $\mathcal{D} = \{\mathbf{x}_i\}_{i=1}^N$ with $N$ datapoints and a pre-trained feature extractor $E_\phi$ parameterized by $\phi$. The pre-trained feature extractor is used to extract embedded representations $\mathbf{h}_i = E_\phi(\mathbf{x}_i)$. Next, a set of nearest neighbors $\mathcal{A}_i$, with cardinality $k$, is computed using the cosine similarity in the embedded representation space. The IC-GAN generator network, $G_\psi$, parameterized by $\psi$, takes as input an embedded representation $\mathbf{h}$ together with a Gaussian noise vector $\mathbf{z} \sim \mathcal{N}(0, I)$, and generates a synthetic image $\tilde{\mathbf{x}} = G_\psi(\mathbf{z}, \mathbf{h})$. IC-GAN is trained using a standard adversarial game between a generator $G_\psi$ and a discriminator $D_\omega$, parameterized by $\omega$, as follows:

$$\min_\psi \max_\omega \; \mathbb{E}_{\mathbf{x}_i \sim p(\mathbf{x}), \mathbf{x}_j \sim \mathcal{A}_i} \left[ \ln D_\omega(\mathbf{x}_j, \mathbf{h}_i) \right] \; + \; \mathbb{E}_{\mathbf{x}_i \sim p(\mathbf{x}), \mathbf{z} \sim \mathcal{N}(0, I)} \left[ \ln(1 - D_\omega(G_\psi(\mathbf{z}, \mathbf{h}_i), \mathbf{h}_i)) \right]. \tag{1}$$

The discriminator $D_\omega$ attempts to distinguish between real samples in $\mathcal{A}_i$ and the generated samples, while the generator $G_\psi$ tries to fool the discriminator by generating realistic images following the distribution of the nearest neighbor samples in $\mathcal{A}_i$. In the class-conditional version of IC-GAN, referred to as CC-IC-GAN, a class label $y$ is used as an extra input conditioning for the generator, such that $\tilde{\mathbf{x}} = G_\psi(\mathbf{z}, \mathbf{h}, y)$; this enables control over the generations given both a class label and an input image.

### 3.2  Data augmentation with IC-GAN

**Data augmentation notation.** We define a data augmentation recipe as a transformation, $T$, of a datapoint, $\mathbf{x}_i \in \mathcal{D}$, to produce a perturbed version $\mathbf{x}_i' = T(\mathbf{x}_i)$ of it. The data augmentation mapping $T : \mathbb{R}^{3 \times H \times W} \to \mathbb{R}^{3 \times H \times W}$ is usually composed of multiple single transformation functions defined in the same domain, $\tau : \mathbb{R}^{3 \times H \times W} \to \mathbb{R}^{3 \times H \times W}$, $T = \tau_1 \circ \tau_2 \circ \dots \circ \tau_t$. Each function $\tau$ corresponds to a specific augmentation of the input such as zooming or color jittering, and is applied with a probability $p_\tau$. Moreover, $\tau$ can be modified with other hyper-parameters $\lambda_\tau$ that are augmentation-specific – e.g. magnitude of zooming, or intensity of color distortion.

**DA$_{\text{IC-GAN}}$.** We introduce a new data augmentation, DA$_{\text{IC-GAN}}$, that leverages a pre-trained IC-GAN generator model and can be used in conjunction with other data augmentation techniques to train a neural network. DA$_{\text{IC-GAN}}$ is applied before any other data augmentation technique and is regulated by a hyper-parameter $p_G$ controlling a percentage of datapoints to be augmented. When a datapoint $\mathbf{x}_i$ is IC-GAN-augmented, it is substituted by the model sample $\tilde{\mathbf{x}}_i = G_\psi(\mathbf{z}, E_\phi(\mathbf{x}_i))$, with $\mathbf{z}$ a Gaussian noise vector. $\tilde{\mathbf{x}}_i$ may then be further transformed with a sequence of subsequent transformations $\tilde{T} = \tau_1 \circ \tau_2 \circ \ldots \circ \tau_{\tilde{t}}$. Note that $\tilde{T}$ might differ from the sequence of transformations $T$ applied when $\mathbf{x}_i$ is not IC-GAN-augmented. We depict this scenario in Figure 2. Moreover, we use the *truncation trick* (Marchesi, 2017) and introduce a second hyper-parameter, $\sigma$, to control the variance of the latent variable $\mathbf{z}$. During IC-GAN training truncation is not applied, and $\mathbf{z}$ is sampled from the unit Gaussian distribution. DA$_{\text{IC-GAN}}$ augmentations can be applied to both supervised and self-supervised representation learning off-the-shelf, see section 4 for details.

## 4 Experimental Setup

In our empirical analysis, we investigate the effectiveness of DA$_{\text{IC-GAN}}$ in supervised and self-supervised representation learning. In the following subsections, we describe the experimental details of both scenarios.

### 4.1 Models, metrics, and datasets

**Models.** For supervised learning, we train ResNets (He et al., 2016) with different depths: 50, 101 and 152 layers, and widths: ResNet-50 twice wider (ResNet-50W2) (Zagoruyko & Komodakis, 2016), and DeiT-B (Touvron et al., 2021). For self-supervised learning, we train the SwAV (Caron et al., 2020) model with a ResNet-50 backbone. For DA$_{\text{IC-GAN}}$, we employ two pre-trained generative models on ImageNet: IC-GAN and CC-IC-GAN, both using the BigGAN (Brock et al., 2019) backbone architecture. IC-GAN conditions the generation process on instance feature representations, obtained with a pre-trained SwAV model[1], while CC-IC-GAN conditions the generation process on both the instance representation obtained with a ResNet-50 trained for classification[2] and a class label. Unless specified otherwise, our models use the default IC-GAN and CC-IC-GAN configuration from Casanova et al. (2021): neighborhood size of $k$=50 and $256 \times 256$ image resolution, trained using only horizontal flips as data augmentation[3]. To guarantee a better quality of generations we set truncation $\sigma = 0.8$ and 1.0 for IC-GAN and CC-IC-GAN respectively. For simplicity, we will use the term (CC-)IC-GAN to refer to both pre-trained models hereinafter.

**Datasets.** We train all the considered models from scratch on ImageNet (IN) (Deng et al., 2009) and test them on the IN validation set. Additionally, in the supervised learning case, models are tested for robustness on a plethora of datasets, including Fake-IN: containing 50K generated images obtained by conditioning the IC-GAN model on the IN validation set; Fake-IN$_{\text{CC}}$: containing 50K images generated with the CC-IC-GAN conditioned on the IN validation set[4]; IN-Adversarial (IN-A) (Hendrycks et al., 2021b): composed of ResNet's adversarial examples present in IN[5]; IN-Rendition (IN-R) (Hendrycks et al., 2021a): containing stylized images such as cartoons and paintings belonging to IN classes; IN-ReaL (Beyer et al., 2020): a relabeled version of the IN validation with multiple labels per image; and ObjectNet (Barbu et al., 2019): containing object-centric images specifically selected to increase variance in viewpoint, background and rotation w.r.t. IN[6]. We also consider the following datasets to study invariances in the learned representations: IN validation set to analyze *instance+viewpoint* invariances; Pascal-3D+ (P3D) (Xiang et al., 2014), composed of ~36K images from 12 categories to measure *instance*, and *instance+viewpoint* invariances; GOT (Huang et al., 2019), 10K video clips with a single moving object to measure invariance to object *occlusion*; and ALOI (Geusebroek et al., 2005), single-object images from 1K object categories with

---

[1] https://dl.fbaipublicfiles.com/deepcluster/swav_800ep_pretrain.pth.tar
[2] https://download.pytorch.org/models/resnet50-19c8e357.pth
[3] https://github.com/facebookresearch/ic_gan
[4] To avoid as much as possible unrealistic generations in creating Fake-IN and Fake-IN$_{\text{CC}}$, for each IN image we generate a set of 20 samples, from which we chose the one most similar (cosine similarity) to the conditioning image in the feature space.
[5] Although IN-A contains samples from only 200 out of the 1000 classes of IN, we compute the results without restricting the predictions to those 200 classes.
[6] The class mapping from ObjectNet to IN is one-to-multi – i.e., one class is mapped to one or more classes of IN. We consider predictions pointing to any of the mapped classes as correct.

plain dark background to measure invariance w.r.t. *viewpoint* (72 viewpoints per object), *illumination color* (12 color variations per object), and *illumination direction* (24 directions per object).

**Metrics.**  We quantify performance for classification tasks as the top-1 accuracy on a given dataset. Moreover, we analyze invariances of the learned representations by using the top-25 Representation Invariance Score (RIS) proposed by Purushwalkam & Gupta (2020). In particular, given a class $y$, we sample a set of object images $\mathcal{T}$ by applying a transformation $\tau$ with different parameters $\lambda_\tau$ such that $\mathcal{T} = \{\tau(x, \lambda_\tau) | \forall \lambda_\tau\}$. We then compute the mean invariance on the transformation $\tau$ of all the objects belonging to $y$ as the average firing rate of the (top-25) most frequently activating neurons/features in the learned representations. We follow the recipe suggested in Purushwalkam & Gupta (2020) and compute the top-25 RIS only for ResNets models, extracting the learned representations from the last ResNet block (2048-$d$ vectors).

**Per-class metrics.**  We further stratify the results by providing class-wise accuracies and correlating them with the quality of the generated images obtained with (CC-)IC-GAN. We quantify the quality and diversity of generations using the Fréchet Inception Distance (FID) (Heusel et al., 2017). We compute per-class FID by using the training samples of each class both as the reference and as the conditioning to generate the same number of synthetic images. We also measure a particular characteristic of the IC-GAN model: the *NN corruption*, which measures the percentage of images in each datapoint's neighborhood that has a different class than the datapoint itself; this metric is averaged for all datapoints in a given class to obtain per-class NN corruption.

## 4.2 Training recipes

In this subsection, we define the training recipe for each model in both supervised and self-supervised learning; we describe which data augmentation techniques are used, how DA$_{\text{IC-GAN}}$ is integrated and the hyper-parameters used to train the models.

**Model selection.**  In all settings, hyper-parameter search was performed with a restricted grid-search for the learning rate, weight decay, number of epochs, and DA$_{\text{IC-GAN}}$ probability $p_G$, selecting the model with the best accuracy on IN validation.

### 4.2.1 Supervised learning

For the ResNet models, we follow the standard procedure in Torchvision[7] and apply random horizontal flips (Hflip) with 50% probability as well as random resized crops (RRCrop) (Krizhevsky et al., 2012). We train ResNet models for 105 epochs, following the setup in VISSL (Goyal et al., 2021). For DeiT-B we follow the experimental setup from Touvron et al. (2021), whose data augmentation recipe is composed of Hflip, RRCrop, RandAugment (Cubuk et al., 2020), as well as color jittering (CJ) and random erasing (RE) (Zhong et al., 2020) —we refer to RandAugment + CJ + RE as RAug—, and typical combinations of CutMix (Yun et al., 2019) and MixUp (Zhang et al., 2017), namely CutMixUp. DeiT models are trained for the standard 300 epochs (Touvron et al., 2021) except when using only Hflip or RRCrop as data augmentation, where we reduce the training time to 100 epochs to mitigate overfitting. Both ResNets and DeiT-B are trained with default hyper-parameters; despite performing a small grid search, better hyper-parameters were not found. Additional details can be found in Appendix A.

**Soft labels.**  We note that the classes in the neighborhoods used to train the (CC-)IC-GAN models are not homogeneous: a neighborhood, computed via cosine similarity between embedded images in a feature space, might contain images depicting different classes. Therefore, (CC-)IC-GAN samples are likely to follow the class distribution in the conditioning instance neighborhood, generating images that may mismatch with the class label from the conditioning image. To account for this mismatch when using (CC-)IC-GAN samples for training, we employ *soft labels*, which are soft class membership distributions corresponding to each instance-specific neighborhood class distribution. More formally, considering the $i$-th datapoint, its $k$-size

---

[7]See the `IMAGENET1K_V1` recipe at https://github.com/pytorch/vision/tree/main/references/classification.

neighborhood in the feature space, $\mathcal{A}_i$, and its class label $y_i \in C$ one-hot encoded with the vector $\mathbf{y}_i$, we compute its soft label as:

$$\mathbf{y}_i^{\text{soft}} = \frac{1}{k} \sum_{j \in \mathcal{A}_i} \mathbf{y}_j, \quad \text{with} \quad \mathbf{y}_j \in \{0,1\}^C \quad \text{and} \quad \sum_c \mathbf{y}_{j,c} = 1. \tag{2}$$

### 4.2.2 Self-supervised learning

We devise a straightforward use of DA$_{\text{IC-GAN}}$ for *multi-view* SSL approaches. Although we chose SwAV (Caron et al., 2020) to perform our experiments, DA$_{\text{IC-GAN}}$ could also be applied to other state-of-the-art methods for *multi-view* SSL like MoCov3 (Chen et al., 2021), SimCLRv2 (Chen et al., 2020b), DINO (Caron et al., 2021) or BYOL (Grill et al., 2020). In this family of approaches, two or more views of the same instance are needed in order to learn meaningful representations. These methods construct multi-view positive pairs $(\mathbf{x}_i', \mathbf{x}_i'')^+$ from an image $\mathbf{x}_i$ by applying two independently sampled transformations to obtain $\mathbf{x}_i' = T(\mathbf{x}_i)$ and similarly for $\mathbf{x}_i''$ (see orange branch in Figure 2). To integrate DA$_{\text{IC-GAN}}$ in such pipelines as an alternative form of data augmentation, we replace $\mathbf{x}_i'$ with a generated image $\tilde{\mathbf{x}}_i'$ with probability $p_G$. To this end, we sample an image $\tilde{\mathbf{x}}_i$ from IC-GAN conditioned on $\mathbf{x}_i$, and apply further hand-crafted data augmentations $\tilde{T}$ to obtain $\tilde{\mathbf{x}}_i' = \tilde{T}(\tilde{\mathbf{x}}_i)$.

**SwAV pre-training and evaluation.** We follow the SwAV pre-training recipe proposed in Caron et al. (2020). This recipe comprises the use of random horizontal flipping, random crops, color distortion, and Gaussian blurring for the creation of each image view. In particular, we investigate two augmentation recipes, differing in the use of the *multi-crop* augmentation (Caron et al., 2020) or the absence thereof. The multi-crop technique augments positive pairs $(\mathbf{x}_i', \mathbf{x}_i'')^+$ with multiple other views obtained from smaller crops of the original image: $(\mathbf{x}_i', \mathbf{x}_i'', \mathbf{x}_i^{\text{small}'''}, \mathbf{x}_i^{\text{small}''''}, ...)^+$. In all experiments, we pre-train SwAV for 200 epochs using the hyper-parameter settings of Caron et al. (2020). To evaluate the learned representation we freeze the ResNet-50 SwAV-backbone and substitute the SSL SwAV head with a linear classification head, which we train supervised on IN validation set for 28 epochs with Momentum SGD and step learning rate scheduler –following the VISSL setup (Goyal et al., 2021).

**Neighborhood augmented SwAV.** To further evaluate the impact of DA$_{\text{IC-GAN}}$ in SSL, we devise an additional baseline, denoted as SwAV-NN, that uses real image neighbors as augmented samples instead of IC-GAN generations: $(\mathbf{x}_j', \mathbf{x}_i'')^+$, $\mathbf{x}_j \in \mathcal{A}_i$. SwAV-NN is inspired by NNCLR (Dwibedi et al., 2021), with the main difference that neighbor images are computed off-line on the whole dataset rather than online using a subset (queue) of the dataset. The nearest neighbors are computed using cosine similarity in the same representation space used for IC-GAN training. With a probability $p_G$, each image in a batch is paired with a uniformly sampled neighbor in each corresponding neighborhood.

## 5 Experimental Evaluation

In this section, we first present the results obtained in the supervised setting using ResNets and DeiT: in-distribution evaluation on IN (Section 5.1.1); classification results on robustness benchmarks (Section 5.1.2); invariance of learned representations (Section 5.1.3); stratified per-class analysis (Section 5.1.4); and sensitivity and ablation studies (Section 5.1.5). Secondly, we show the SSL results of SwAV on IN (Section 5.2).

### 5.1 Supervised ImageNet training

### 5.1.1 In-distribution evaluation

We start by analyzing the impact of DA$_{\text{IC-GAN}}$ when used in addition to several hand-crafted data augmentation recipes for ResNet-50, ResNet-101, ResNet-152, ResNet-50W2, and DeiT-B. In Figure 3, we report the top-1 accuracy on the IN validation set for the models under study.

When training ResNets (see Figures 3a-3d), coupling DA$_{\text{IC-GAN}}$ with random horizontal flips (Hflip) results in an overall accuracy boost of 0.5-1.7%p when using IC-GAN and 0.3-1.6%p when using CC-IC-GAN. How-

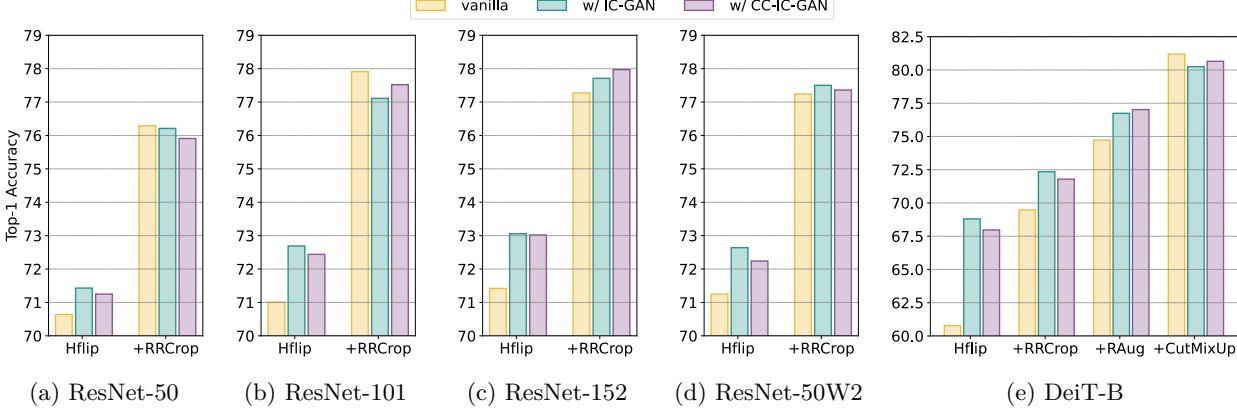

Figure 3: Impact of $\text{DA}_{\text{IC-GAN}}$ when coupled with different data augmentation (DA) recipes for training on IN. Hand-crafted DA techniques are: Hflip –random horizontal flips–, RRCrop –random resized crops–, RAug –RandAugment(Cubuk et al., 2020)–, and CutMixUp (Yun et al., 2019; Zhang et al., 2017). DA techniques are added from left to right, with the right-most column combining all possible DA strategies; i.e, +RRCrop applies RRCrop on top of Hflip.

ever, when pairing $\text{DA}_{\text{IC-GAN}}$ with random horizontal flips and crops (+RRCrop), we observe an accuracy decrease of 0.1-0.4%p for ResNet-50 and 0.4-0.8%p for ResNet-101, while for the bigger capacity ResNets the accuracy is boosted by 0.4-0.7%p for ResNet-152 and 0.2%p for ResNet-50W2 with either IC-GAN or CC-IC-GAN. These results show that $\text{DA}_{\text{IC-GAN}}$ is beneficial for higher capacity networks, which might be able to capture the higher image diversity induced by more aggressive DA recipes. These observations align with those of Kolesnikov et al. (2020), who showed that extremely large dataset sizes may be detrimental to low-capacity networks such as ResNet-50 and ResNet-101, and those of Steiner et al. (2021), who showed that hand-crafted DA strategies applied on large-scale datasets can also result in performance drops.

When training DeiT-B (see Figure 3e), the largest model considered –with $\sim 4\times$ as many parameters as the ResNet-50– and which usually needs aggressive regularization strategies (Steiner et al., 2021; Touvron et al., 2022), we observe that $\text{DA}_{\text{IC-GAN}}$ paired with random horizontal flips (Hflip), random crops (+RRCrop) and RandAugment (+RAug) provides a remarkable accuracy boost of 8.0/7.2%p, 2.9/2.3%p, 2.0/2.3%p for IC-GAN and CC-IC-GAN, respectively, when compared to only using the hand-crafted DA recipes. However, when extending the recipe by adding more aggressive DA such as CutMixUp, the combination with $\text{DA}_{\text{IC-GAN}}$ results in a slight decrease of -0.5%p and -1.0%p in accuracy for CC-IC-GAN and IC-GAN respectively.

Overall, $\text{DA}_{\text{IC-GAN}}$ boosts the top-1 accuracy when paired with most of the hand-crafted DA recipes studied and with larger ResNet models, showcasing the promising application of $\text{DA}_{\text{IC-GAN}}$ as a DA tool. We hypothesize that $\text{DA}_{\text{IC-GAN}}$ acts as an implicit regularizer and as such, when paired with the most aggressive DA recipes for smaller ResNet models and DeiT-B, does not lead to an accuracy improvement, possibly due to an over-regularization of the models. Moreover, we argue that state-of-the-art training recipes with hand-crafted DA strategies have been carefully tuned and, therefore, simply adding $\text{DA}_{\text{IC-GAN}}$ into the mix without careful tuning of training and hand-crafted DA hyper-parameters or the optimization strategy might explain the decrease in accuracy for these recipes.

### 5.1.2 Robustness evaluation

We present results on six additional datasets: Fake-IN, Fake-IN$_{\text{CC}}$, IN-A, IN-R, IN-Real and ObjectNet, to test the robustness of our models. We consider the ResNet-50 model for its ubiquitous use in the literature, as well as the high capacity models ResNet-152 and DeiT-B for their high performance. Results are reported in Table 1.

On Fake-IN and Fake-IN$_{\text{CC}}$, datasets composed of generated images obtained with IC-GAN and CC-IC-GAN respectively, we make two observations. First, the decrease in accuracy of the vanilla ResNets and DeiT-B

Table 1: Robustness evaluation. Top-1 accuracy for ResNet-50, ResNet-152 and DeiT-B, trained on IN and evaluated on: IN-Real (IN-ReaL), Fake-IN, FAKE-IN$_{CC}$, IN-A(IN-A), IN-R (In-R) and ObjectNet. IN (in distribution) results are reported for reference.

| | | | | robustness benchmarks | | | | | |
| Model | DA base | DA$_{IC\text{-}GAN}$ | IN | IN-ReaL | Fake-IN | Fake-IN$_{CC}$ | IN-A | IN-R | ObjectNet |
|---|---|---|---|---|---|---|---|---|---|
| ResNet-50 | Hflip | / | 70.75 | 74.18 | 33.11 | 55.00 | 2.07 | **23.07** | **33.33** |
| | | w/ IC-GAN | **71.43** | 74.38 | **39.53** | **58.06** | 0.97 | 21.46 | 31.93 |
| | | w/ CC-IC-GAN | 71.25 | **74.63** | 33.38 | 57.70 | **2.23** | 22.47 | **33.32** |
| | + RRCrop | / | **76.29** | **77.52** | 37.55 | 61.87 | **0.61** | **23.28** | **34.67** |
| | | w/ IC-GAN | 76.21 | 77.23 | **40.65** | 63.14 | 0.45 | 22.99 | 34.45 |
| | | w/ CC-IC-GAN | 75.91 | 77.21 | 38.55 | **65.06** | 0.60 | 22.70 | 33.52 |
| ResNet-152 | Hflip | / | 71.42 | 73.90 | 34.85 | 58.61 | 1.68 | 23.02 | 33.44 |
| | | w/ IC-GAN | **73.06** | **75.29** | **38.28** | 60.52 | **2.31** | 24.24 | **35.44** |
| | | w/ CC-IC-GAN | 73.02 | 75.09 | 34.39 | **65.02** | 2.08 | **24.93** | 35.28 |
| | + RRCrop | / | 77.27 | 78.17 | 37.88 | 63.64 | 1.56 | 24.68 | 35.51 |
| | | w/ IC-GAN | 77.71 | **78.90** | **40.56** | 64.11 | 2.32 | **26.03** | 38.16 |
| | | w/ CC-IC-GAN | **77.97** | 78.87 | 37.90 | **66.50** | **2.57** | 25.96 | **38.27** |
| DeiT-B | Hflip + RRCrop | / | 69.47 | 70.46 | 35.78 | 59.38 | 1.85 | 15.82 | 20.36 |
| | | w/ IC-GAN | **72.35** | **73.59** | **43.79** | 62.63 | 1.92 | 18.36 | **24.49** |
| | | w/ CC-IC-GAN | 71.79 | 72.74 | 36.41 | **75.35** | **2.12** | **18.74** | 23.78 |
| | + RAug | / | 75.28 | 75.56 | 34.53 | 59.75 | 4.36 | 24.18 | 27.31 |
| | | w/ IC-GAN | 76.74 | 77.37 | **42.80** | 64.49 | 4.34 | 25.11 | **31.85** |
| | | w/ CC-IC-GAN | **77.02** | **77.53** | 37.07 | **75.71** | **4.86** | **26.37** | 31.49 |
| | + CutMixUp | / | **81.19** | **81.23** | 37.87 | 66.90 | **11.95** | 31.63 | **40.56** |
| | | w/ IC-GAN | 80.16 | 80.84 | **41.78** | 67.33 | 11.14 | 30.85 | 38.59 |
| | | w/ CC-IC-GAN | 80.65 | 80.97 | 38.23 | **76.41** | 11.70 | **32.23** | 38.58 |

on these datasets with respect to their IN accuracy highlights a considerable data distribution shift between IN and both Fake-IN and Fake-IN$_{CC}$. Moreover, the accuracy on Fake-IN is significantly lower than on Fake-IN$_{CC}$, as one may expect given the higher generation quality and label preservation of CC-IC-GAN. Secondly, the use of IC-GAN and CC-IC-GAN provides significant boosts on the respective Fake-IN and Fake-IN$_{CC}$ datasets, highlighting the increased robustness of the models trained with DA$_{IC\text{-}GAN}$ while remaining competitive on IN.

On IN-A and IN-R, DA$_{IC\text{-}GAN}$ outperforms the vanilla baselines in most of the settings explored, while especially increasing robustness for larger models – i.e., ResNet-152 and DeiT-B. However, we notice a better impact of CC-IC-GAN compared to IC-GAN in 6/7 cases for IN-A and in 5/7 for IN-R, which overturns the results on IN validation where IC-GAN is better in most cases. This might be explained by the fact that despite being less diverse, CC-IC-GAN generations are more likely to depict the correct class; during training the lower sample diversity reduces the regularization effect providing lower in-distribution gains.

Finally, on ObjectNet and IN-ReaL, we observe similar trends to those in IN: ResNet-152 and DeiT-B with horizontal flips, random crops and random augment benefit from DA$_{IC\text{-}GAN}$, leading to an increase in accuracy. This evidences that the improvements that DA$_{IC\text{-}GAN}$ provides in-distribution to high capacity models transfer well when considering a more correct IN labeling, such as the one of IN-ReaL, and more importantly when classifying different objects with several viewpoints and backgrounds, such as those in ObjectNet.

Overall, this robustness evaluation confirms a positive impact of DA$_{IC\text{-}GAN}$ for high-capacity models – already benefiting on in-distribution data–, suggesting that they learn more robust representations which may transfer to unseen datasets. In particular, the generations of (CC-)IC-GAN appear to increase the robustness of the trained models by presenting them with slightly different characteristics from in-distribution images.

### 5.1.3   Feature invariances

We study the invariance of the learned representations of the ResNet-152 model–our best-performing ResNet model–, to assess whether the DA$_{IC\text{-}GAN}$'s performance boosts could be attributed to more robust learned representations. In particular, we evaluate representation invariances to *instance*, *viewpoint*, *occlusion*, and *illumination*, in terms of the top-25 RIS scores. Results are reported in Table 2.

Table 2: Top-25 Representation Invariance Score (RIS) of the learned representations, evaluated on ImageNet (IN), Pascal3D (P3D), GOT-10K (GOT), and ALOI datasets. ↑ top-25 RIS means ↑ invariance.

| Model | DA base | DA$_{\text{IC-GAN}}$ | P3D Instance | P3D Inst. + View. | GOT Occlusion | ALOI Viewpoint | ALOI IllumColor | ALOI IllumDir |
|---|---|---|---|---|---|---|---|---|
| ResNet-152 | Hflip | / | 57.05 | 61.15 | **73.12** | **81.23** | **98.91** | **90.11** |
| | | w/ IC-GAN | **58.64** | 61.63 | 70.63 | 78.19 | 98.90 | 87.27 |
| | | w/ CC-IC-GAN | 58.58 | **62.02** | 71.09 | 78.68 | 98.65 | 87.53 |
| | + RRCrop | / | 59.74 | 62.86 | 74.06 | 83.53 | **99.67** | 90.00 |
| | | w/ IC-GAN | **62.22** | **65.87** | **74.37** | 83.89 | 99.63 | **91.31** |
| | | w/ CC-IC-GAN | 62.01 | 64.93 | 74.36 | **84.06** | 99.65 | 91.23 |

We measure the invariance to *instance* on P3D. We observe that DA$_{\text{IC-GAN}}$ always induces a higher RIS. We argue that this result might be expected by considering the ability of (CC-)IC-GAN of populating the neighborhood of each datapoint, i.e., instance.

Next, we quantify the invariance to *viewpoint* using both P3D – *instance + viewpoint* – and ALOI. In this case, we also notice that DA$_{\text{IC-GAN}}$ generally induces more consistent representations, except when combined with horizontal flips and evaluated on ALOI. Our explanation for the generally higher viewpoint invariance is that (CC-)IC-GAN samples depict slightly different viewpoints of the object present in the conditioning image – see visual examples in Appendix C.

Finally, by looking at the *occlusion* invariance on GOT, and the *illumination color* and *direction* invariance on ALOI, we observe mixed results: in some cases DA$_{\text{IC-GAN}}$ slightly increases the RIS while in some other cases DA$_{\text{IC-GAN}}$ slightly decreases it. This result is perhaps unsurprising as none of these invariances is directly targeted by DA$_{\text{IC-GAN}}$; and the slight increases observed in some cases could be a side-effect of the larger diversity given by DA$_{\text{IC-GAN}}$– e.g., higher occlusion invariance might be due to erroneous generations not containing the object class.

Overall, the invariance analysis highlights that DA$_{\text{IC-GAN}}$, by leveraging the diversity of the neighborhood, can be useful not only to regularize the model and achieve better classification accuracy, but also to provide more consistent feature representations across variations of *instance* and *viewpoint*. Guaranteeing such invariances is likely to lead to a better transferability/robustness of the representations – as shown in Section 5.1.2.

### 5.1.4 Per-class analysis

To further characterize the impact of DA$_{\text{IC-GAN}}$, we perform a more in-depth analysis by stratifying the ResNet-152 results per class. We compare the per-class FID of IC-GAN and CC-IC-GAN, as well as their NN corruption, with the top-1 accuracy per class of a vanilla model – trained without DA$_{\text{IC-GAN}}$– and a model trained only with generated samples – i.e., using DA$_{\text{IC-GAN}}$ with $p_G = 1.0$ – with the goal of better understanding the impact of (CC-)IC-GAN's generations on the model's performance. Results are reported in Figure 4. Note that the exclusive use of generated samples leads to rather low top-1 accuracy: $\sim$43% and $\sim$46% for ResNet-152 when using IC-GAN and CC-IC-GAN respectively.

Figures 4a and 4b present the per-class FID of (CC-)IC-GAN as a function of per-class top-1 accuracy of the vanilla baseline and the DA$_{\text{IC-GAN}}$ models. We observe that DA$_{\text{IC-GAN}}$ tends to exhibit higher accuracy for classes with lower FID values, and lower accuracy for classes with higher FID values overall. In particular, classes for which generated images have good quality and diversity (e.g., $\sim$ 50 FID or lower) tend to achieve high top-1 accuracy for both the vanilla model and the one trained with only generated data. Conversely, when the FID of a class is high, its per-class accuracy oftentimes drops for the (CC-)IC-GAN-trained models, whereas the vanilla model remains performant. Perhaps unsurprisingly, this evidences that leveraging image generations of poorly modeled classes to train the ResNet-152 model hurts the performance. Moreover, we note that there are more classes with very high FID ($\sim$ 200 or higher) for IC-GAN than CC-IC-GAN. Intuitively, this could be explained by the fact that CC-IC-GAN uses labels to condition the model and appears to be less prone to mode collapse (see Figure 5).

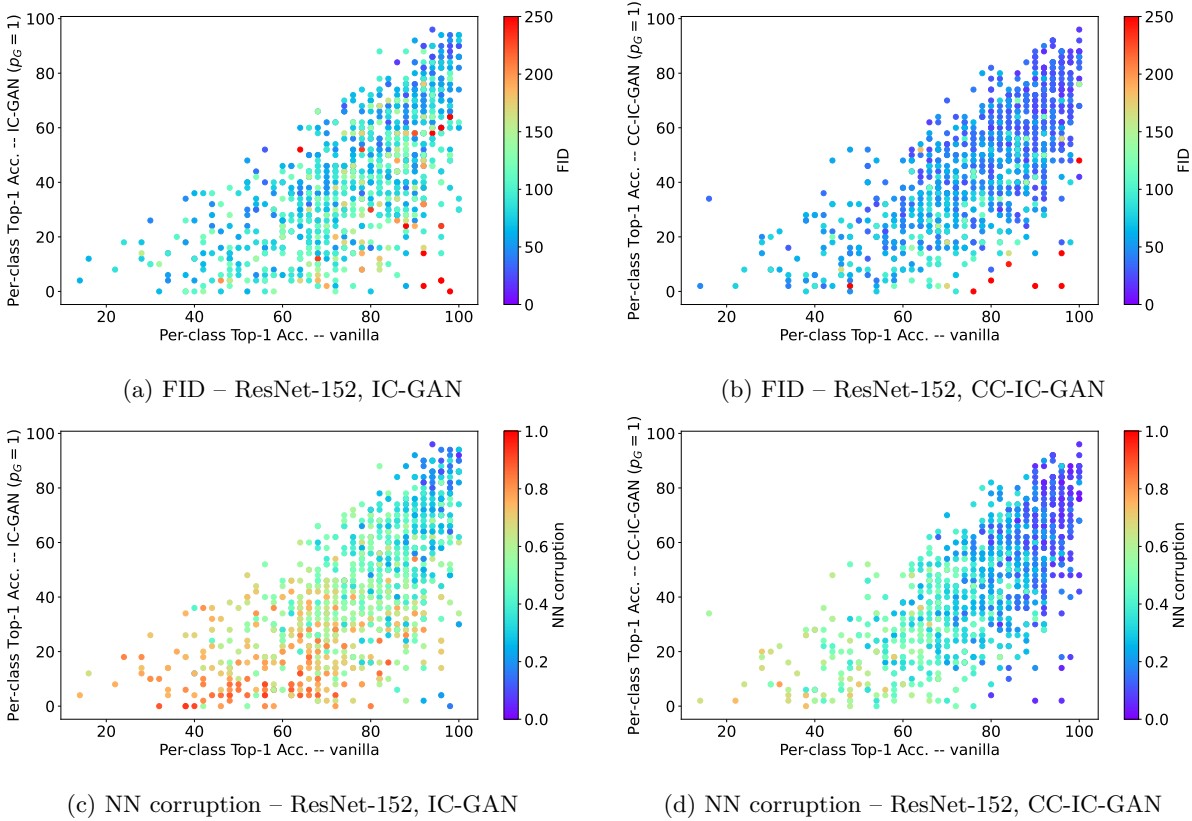

(a) FID – ResNet-152, IC-GAN

(b) FID – ResNet-152, CC-IC-GAN

(c) NN corruption – ResNet-152, IC-GAN

(d) NN corruption – ResNet-152, CC-IC-GAN

Figure 4: Impact of (CC-)IC-GAN's generation quality on per-class performance. (a-b) Per-class FID as a function of per-class top-1 accuracy of the vanilla and $DA_{IC\text{-}GAN}$ models. We observe that higher quality (CC-)IC-GAN generations tend to result in improved performances. (c-d) Per-class NN corruption as a function of per-class top-1 accuarcy of the vanilla and $DA_{IC\text{-}GAN}$ models. We observe that less corrupted classes tend to result in improved performances. ImageNet validation results shown for the ResNet-152 model trained with horizontal flips and random crops. We limited FID colormap interval to 250 to aid interpretability, while we observed FID values up to 500 for certain classes.

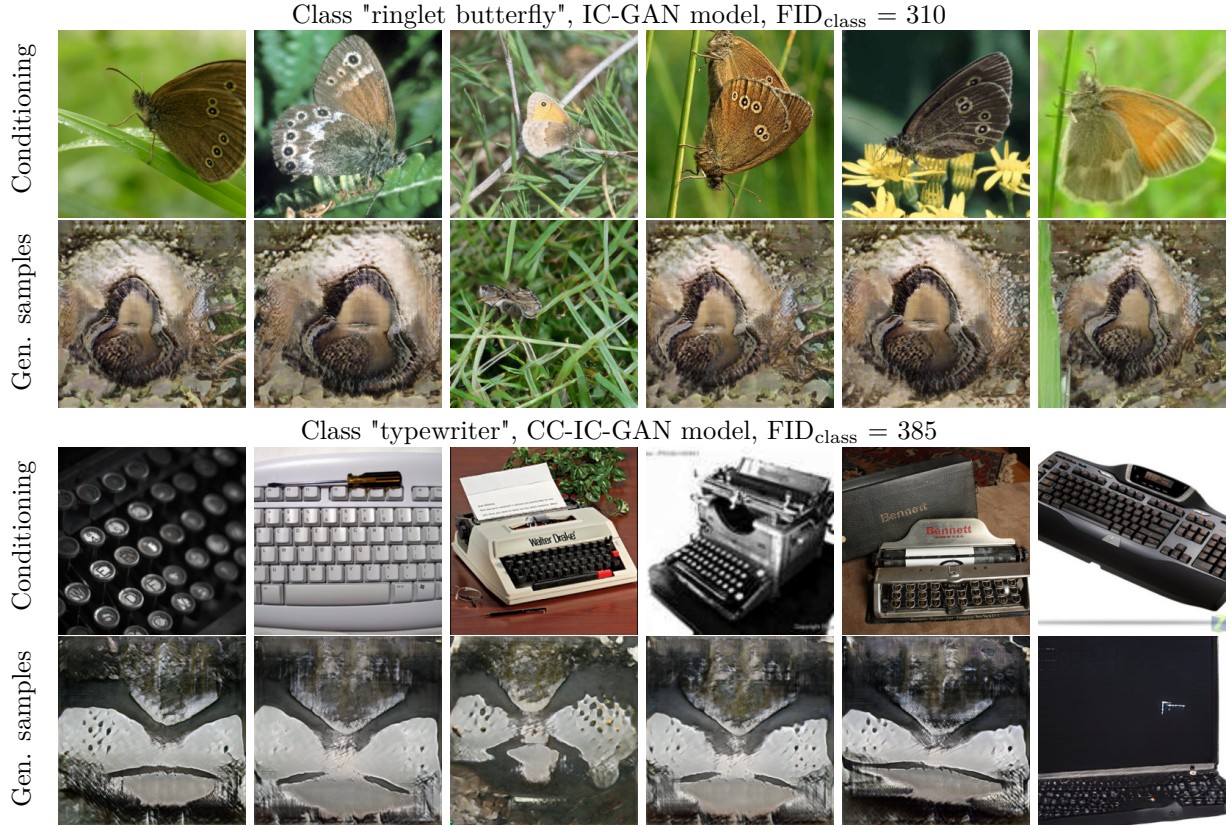

Figure 5: Conditioning sample and one of its generated samples with IC-GAN or CC-IC-GAN, illustrating the mode collapse for some classes in ImageNet. Note that the mode collapse is evidenced in different classes for IC-GAN and CC-IC-GAN.

We additionally observe in Figures 4c and 4d that the low accuracies of the model trained with generated data can be partially explained by the NN corruption: classes with less corrupted neighborhoods tend to exhibit higher top-1 accuracies than the more corrupted ones. However, we observe some specific cases of classes with low corruption which result in very low accuracy when considering the model trained with all generated samples (see the bottom-right corner of the plots). This could be explained by the mode collapse that (CC-)IC-GAN experience, as we see that those same classes generally have very high FID ($> 200$) in Figures 4a and 4b.

In this analysis, we shed some light on the problematic (CC-)IC-GAN modeling of certain classes. We believe that computing stratified results for generative models might be a good practice to be adopted by the community, as also supported by Ravuri & Vinyals (2019). Nevertheless, the observed positive correlation between high classification accuracy and (CC-)IC-GAN's generation quality –studied through the lens of per-class FID and NN corruption– constitutes a promising result to improve the effectiveness of DA$_{\text{IC-GAN}}$. To this end, we ran an additional experiment where we avoid applying DA$_{\text{IC-GAN}}$ on classes having very high FID ($>= 150$), i.e., where (CC-)IC-GAN has very low generation quality. We report the results in Appendix B. Notably, the impact of leveraging DA$_{\text{IC-GAN}}$ could be potentially improved by increasing the generation quality of the (CC-)IC-GAN's poorly modeled classes. These findings improve upon those of Ravuri & Vinyals (2019), where a pre-trained BigGAN showed little to no correlation between FID and classification accuracy in a similar setting, strengthening the position of instance-conditioned models such as (CC-)IC-GAN. We observe similar trends for DeiT-B (see Appendix B).

### 5.1.5 Sensitivity and ablation studies

**Probability $p_G$.** We study the impact of the probability of applying DA$_{\text{IC-GAN}}$, $p_G$, in Figure 6. We consider our best ResNet model as well as DeiT-B. We further include the study on ResNet-50 as a sanity

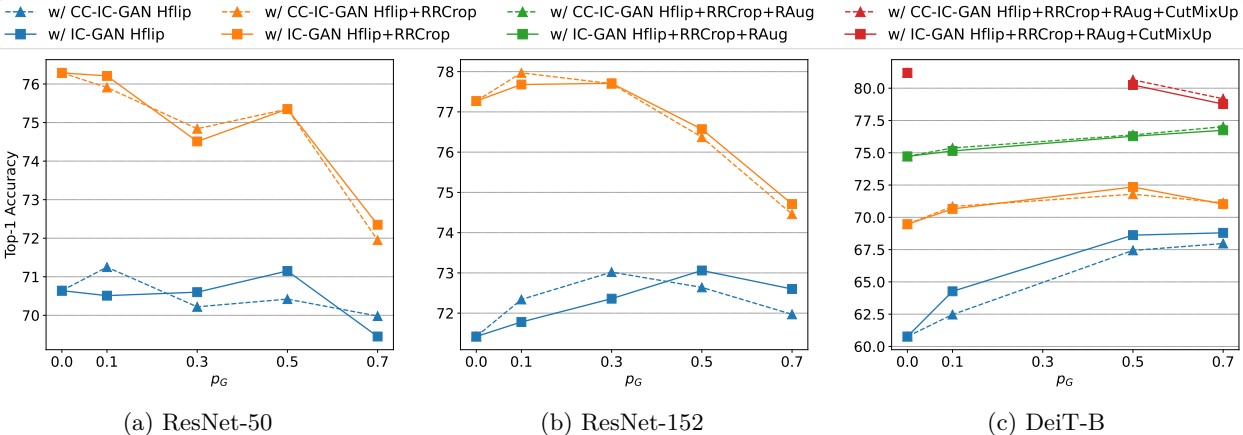

Figure 6: Sensitivity study for the probability $p_G$ of applying DA$_\text{IC-GAN}$. The missing datapoint in red curves of panel (c) are due to trainings not converging.

check to validate our previous over-regularization hypothesis. When coupling DA$_\text{IC-GAN}$ with horizontal flip to train the ResNet-50 model (Figure 6a), we observe that a probability of $p_G = 0.1$ and $p_G = 0.5$ achieve the best results for CC-IC-GAN and IC-GAN, respectively. However, when adding random crops to the recipe, ResNet-50 no longer benefits from DA$_\text{IC-GAN}$ and obtains the best results for $p_G = 0$, highlighting the potential over-regularization suffered by low-capacity models as discussed in section 5.1.1. When it comes to ResNet-152 (Figure 6b), we observe that the overall accuracy increases until achieving its peak value for some $p_G$ and then starts decreasing. More precisely, CC-IC-GAN shows optimal $p_G$ for lower values, 0.1 and 0.3, whereas IC-GAN benefits from the higher probability values 0.3 and 0.5. Note that in both cases, DA$_\text{IC-GAN}$ coupled with random horizontal flips and crops requires lower $p_G$ values than DA$_\text{IC-GAN}$ coupled with horizontal flips only, emphasizing the benefit of DA$_\text{IC-GAN}$ especially when leveraging soft augmentation strategies. For DeiT-B architecture (Figure 6c), we note that increasing $p_G$ values mostly result in better accuracy when using all DA recipes except the strongest one containing CutMixUp. This trend might be due to the higher capacity of the DeiT-B model that combined with the lower architectural inductive bias – i.e., no convolution – requires stronger regularization on IN. This shows the benefits of using DA$_\text{IC-GAN}$ to regularize training, especially for architectures prone to overfitting, which require higher $p_G$ values.

**CutMixUp components.** In Table 3 we present an ablation of the two components in CutMixUp – CutMix and MixUp – when training DeiT-B models with and without DA$_\text{IC-GAN}$. To facilitate convergence when using CutMix and/or MixUp, we follow Touvron et al. (2022) and perform label smoothing. Moreover, we leverage the *soft labels* introduced in section 5.1 for all experiments using DA$_\text{IC-GAN}$. We observe that removing CutMix results in a decrease in accuracy of 0.9%p, whereas removing MixUP increases the performance by +0.5%p. Removing both CutMix and MixUp results in the lowest accuracy. When considering DA$_\text{IC-GAN}$,

Table 3: CutMixUp ablation when training DeiT-B with Hflip+RRCrop+RAug and DA$_\text{IC-GAN}$. All models trained with a label smoothing of 0.1. *: Failed to converge.

| IC-GAN | CC-IC-GAN | CutMix | MixUp | Top-1 |
|--------|-----------|--------|-------|-------|
| ✗ | ✗ | ✓ | ✓ | 81.2 |
| ✗ | ✗ | ✓ | ✗ | 81.7 |
| ✗ | ✗ | ✗ | ✓ | 80.3 |
| ✗ | ✗ | ✗ | ✗ | 77.4 |
| ✓ | ✗ | ✓ | ✓ | 80.2 |
| ✓ | ✗ | ✓ | ✗ | 78.5 |
| ✓ | ✗ | ✗ | ✓ | * |
| ✓ | ✗ | ✗ | ✗ | 78.2 |
| ✗ | ✓ | ✓ | ✓ | 80.7 |
| ✗ | ✓ | ✓ | ✗ | 78.2 |
| ✗ | ✓ | ✗ | ✓ | * |
| ✗ | ✓ | ✗ | ✗ | 78.1 |

we observe that results are consistently better than those of the baseline model when not using CutMix nor MixUp. Leveraging CutMixUp appears to be beneficial but the induced accuracy gains remain lower than those experienced by the baseline model, suggesting that additional tuning of CutMixUp might be required when coupled with DA$_\text{IC-GAN}$.

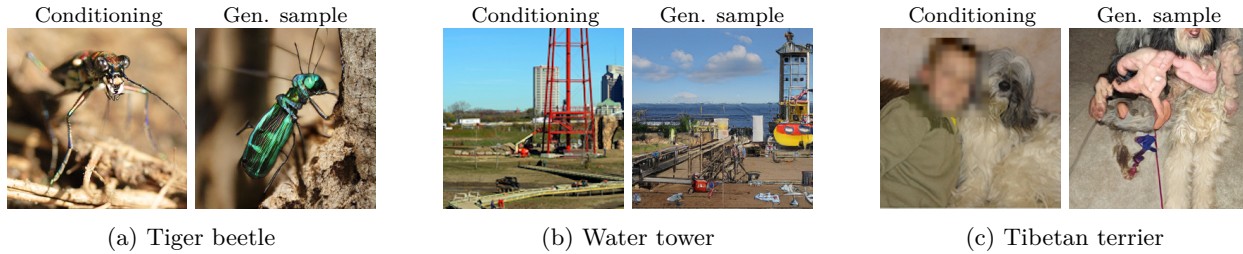

| Conditioning | Gen. sample | Conditioning | Gen. sample | Conditioning | Gen. sample |
| --- | --- | --- | --- | --- | --- |
| (a) Tiger beetle | | (b) Water tower | | (c) Tibetan terrier | |

Figure 7: Example of three IC-GAN generations, one (a) qualitatively and semantically correct, one (b) semantically incorrect, and one (c) with poor quality. Left image in each pair is the conditioning, where the red square shows the central crop actually used by IC-GAN.

## 5.2 Self-supervised ImageNet training

Given that IC-GAN can be trained without labels, and can synthesize images without the need of class categories, we employ it to extend multi-view SSL pre-training on IN (He et al., 2020; Chen et al., 2020a; Caron et al., 2020) by integrating $DA_{IC-GAN}$ into the standard hand-crafted DA recipes. In Table 4 we report the accuracy scores obtained on the IN validation set when testing the pre-trained SSL methods via linear classifier evaluation.

We observe a clear difference between the two SwAV settings: (i) with single-crop, and (ii) with multi-crop. In the former case, the use of $DA_{IC-GAN}$ boosts the top-1 classification accuracy of the linear evaluation probe by ∼1%p. On the contrary, when $DA_{IC-GAN}$ is combined with multi-crop RRC (crops with different sizes and zooms of the original image), we observe a detrimental effect, with roughly a 2%p accuracy drop. It is worth noting that the multi-crop transformation already results in significant variations from the original image, in some cases even changing its semantics. Hence, combining $DA_{IC-GAN}$ and multi-crop might result in extreme augmentation diversity (see example reported in Appendix C). Moreover, we recall that the SwAV model has a

Table 4: SwAV accuracy on ImageNet validation using different image sources and DA methods. For each view (view1 and view2) in the multi-view SSL setup, the *image source* can be: *Original* - real image from the dataset, or *NN* - a randomly sampled neighbor (k-NN with k=50). $DA_{IC-GAN}$ is applied on top of the image source. *RRC*: RandomResizedCrop + ColorDistortion + Gaussian Blurring. RRC can produce a *single*-crop or *multi*-crop.

| | View1 | | View2 | | Top-1 |
| --- | --- | --- | --- | --- | --- |
| Image source | $DA_{IC-GAN}$ | RRC | Image source | RRC | |
| Original | ✗ | single | Original | single | 67.96 |
| Original | ✓ | - | Original | single | 68.90 |
| NN | ✗ | single | Original | single | **70.06** |
| Original | ✗ | single | Original | multi | 73.64 |
| Original | ✓ | single | Original | multi | 71.72 |
| NN | ✗ | single | Original | multi | **73.73** |

ResNet-50 backbone, whose capacity might be too low to capture such a large image diversity, as discussed in the supervised IN training in Section 5.1. A further confirmation of this hypothesis may come from the results of SwAV-NN, which are positive for single-crop, with a ∼ 2%p increase, while remaining on-par for multi-crop (+0.1%); using real image neighbors instead of IC-GAN-generated ones (last row of Table 4) does not increase the diversity in the training distribution, requiring less model capacity. Moreover, a non-negligible role might be played by the quality of (CC-)IC-GAN generations that for some instances (or classes) might be poor or semantically far from the conditioning, as observed in Section 5.1.4 and visually shown in Figure 7.

Overall, this empirical analysis reveals that $DA_{IC-GAN}$ improves SSL training only when single-crop augmentation is adopted, while the use of stronger hand-crafted DA (multi-crop) on top of IC-GAN-generated images is detrimental. We point out that in NNCLR (Dwibedi et al., 2021) no gains from the combination of the multi-crop augmentation with the neighbor-based augmentation were found, and our SwAV-NN experiments confirm this finding.

## 6 Conclusions

We have studied the potential of Instance-Conditioned GAN (IC-GAN) as a data augmentation technique in state-of-the-art training recipes for visual representation learning. Specifically, we have presented $\text{DA}_{\text{IC-GAN}}$, a data augmentation module which leverages the generations of (CC-)IC-GAN and integrates them seamlessly with standard handcrafted data augmentation recipes. We have validated $\text{DA}_{\text{IC-GAN}}$ in the context of image classification, leveraging supervised learning with ResNets (He et al., 2016) and DeiT-B (Touvron et al., 2021), as well as self-supervised learning with SwAV (Caron et al., 2020). The results of this validation have unveiled a beneficial impact of $\text{DA}_{\text{IC-GAN}}$, especially for higher capacity networks and when coupling (CC-)IC-GAN augmentations with soft hand-crafted augmentation strategies, suggesting $\text{DA}_{\text{IC-GAN}}$ may act as an implicit regularizer for the models. Additionally, we have found that the representations learned when training models with $\text{DA}_{\text{IC-GAN}}$ are more robust when transferred to unseen datasets and more invariant across variations in instance and viewpoint, as a byproduct of augmenting the dataset with generated images obtained with (CC-)IC-GAN. Moreover, with a per-class stratification of the results, we have discovered a correlation between per-class performance and generated per-class image quality. These findings hint at two future directions to improve the effect of $\text{DA}_{\text{IC-GAN}}$: increasing generation quality for the classes which are poorly modeled by (CC-)IC-GAN and to tune the (CC-)IC-GAN augmentations per class. Finally, in the case of more aggressive data augmentation techniques for which $\text{DA}_{\text{IC-GAN}}$ does not provide an improvement over the baselines, such as CutMixUp or multi-crop, we hypothesize that those augmentation recipes already result in strong image variations, and consequently, combining those with (CC-)IC-GAN generations out-of-the-box through $\text{DA}_{\text{IC-GAN}}$ may cause an over-regularization of the training.

To conclude, we have shown that current state-of-the-art large capacity models can be improved using instance conditioned generative models such as IC-GAN in conjunction with hand-crafted data augmentation techniques. We further hypothesize that by boosting the quality and diversity of instance conditioned samples, models may eventually stop relying on hand-crafted data augmentation techniques altogether, and instead move towards completely data-driven augmentation schemes to obtain infinitely many realistic augmented samples.

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

## A    Implementation and Practical Tips

### A.1    Data augmentation and optimization hyper-parameters

**Handcrafted data augmentations.**    We used the VISSL library (Goyal et al., 2021) which relies on Torchvision transformations[8], and considered the following data augmentation strategies:

- Hflip: Random horizontal flipping applied with 50% probability.

- RRCrop: Random resized cropping. The location of the crop is sampled uniformly based on the sampled crop size, which is in the range (0.08, 1). When adopted in our experiments, we apply it to all images in a batch.

- RAug = RandAugment + Color Jittering (CJ) + Random erasing (RE): When enabled, RandAugment (Cubuk et al., 2020) is applied to all images with magnitude $9.0 \pm 0.5$ and increasing distortion severity for higher magnitude values. At each iteration RandAugment randomly chooses two types of distortions. CJ distorts brightness, contrast, saturation, and hue, each with probability 0.4. RE is applied with probability 0.25, erasing a rectangle of size sampled from (0.02, 0.33).

- CutMixUp. CutMix (Yun et al., 2019) and MixUp (Zhang et al., 2017) are never applied simultaneously; there is a 0.5 probability of choosing one or the other at each iteration. Note that Mixup is applied 80% of the time when selected. As previously mentioned, we adopt label smoothing of 0.1 to ease convergence when using CutMixUp.

---

[8] https://pytorch.org/vision/stable/transforms.html

**Optimization hyper-parameters.** In Table 5, we list the optimization hyper-parameters explored for supervised training on ImageNet. Note that for all supervised experiments, we optimized the multi-class cross-entropy loss.

Table 5: Training hyper-parameters for supervised training on ImageNet.

| Model | Optimizer | Epochs | Learning rate (LR) | LR scheduler | LR scaling | Weight decay |
|---|---|---|---|---|---|---|
| ResNet-50 | Mom. SGD | 105 | {(5, 2, 1)e-1, (5, 1)e-2, 5e-3} | Step(30,60,90,100) | Lin-256 | {(1, 5)e-5, (1, 5)e-4, 1e-3} |
| ResNet(-101, -152, 50W2) | Mom. SGD | 105 | 1e-1 | Step(30,60,90,100) | Lin-256 | 1e-4 |
| DeiT-B | AdamW | 100/300 | 1/5e-4 | Lin + Cosine | Lin-512 | 1e-1/5e-2 |

## A.2 Implementation details

**Fixing the number of IC-GAN-augmented datapoints.** Variable batch size can cause unexpected breaks in GPU-accelerated computations, mostly due to GPU memory pre-allocation. To avoid this phenomenon, we fix the number of IC-GAN-augmented images in a batch to be `ceil(batch_size * p_G)`.

**Computational overhead of DA$_{\text{IC-GAN}}$.** Adding DA$_{\text{IC-GAN}}$ to the training recipe requires some additional space and time for the IC-GAN generation. For instance, in terms of space, $\sim$11GB of a single GPU memory are required to generate a batch of 64 images at $256 \times 256$ resolution. In terms of time, we noticed that training ResNets with DA$_{\text{IC-GAN}}$ and $p_G = 1.0$ doubles the training time, whereas for $p_G = 0.5$ the time requirement increases by roughly 50%. However, we did not take advantage of half-precision computations nor of any other inference-only trick like `jit` scripting in PyTorch. We hypothesize that exploring such optimizations might significantly reduce the computational overhead of DA$_{\text{IC-GAN}}$.

**Pre-computing dataset embeddings.** The IC-GAN generation step requires feature representation of the conditioning images. In order to reduce the computation needed during the training, we compute the embeddings of the entire training dataset in advance, and store them in an array which is loaded into memory at the beginning of the training.

**Hardware used for experiments.** For most of the experiments, we performed distributed training using cluster nodes with 8 Nvidia V100 GPUs with 32GB memory. We changed the number of nodes based on the training model and the desired batch size – e.g., 1 node for ResNets, 4 for DeiT-B, and 8 for SwAV.

# B Additional Results

## B.1 DeiT-B per-class analysis

Figure 8 assesses the impact of (CC-)IC-GAN's generation quality on the per class performance of the DeiT-B model. The exclusive use of generated samples to train DeiT-B leads to rather a low top-1 accuracy of $\sim$48% and $\sim$51%, when using IC-GAN and CC-IC-GAN respectively.

Following section 5.1.4, Figures 8 (a–b) show the per-class FID of (CC-)IC-GAN as a function of per-class top-1 accuracy of the vanilla baseline and the DA$_{\text{IC-GAN}}$ models. We observe similar trends as for the ResNet-152 models, – i.e., DA$_{\text{IC-GAN}}$ tends to exhibit higher accuracy for classes with lower (better) FID values, and lower accuracy for classes with higher FID values, suggesting that using image generations of poorly modeled classes hurts the performance of DeiT-B. Figures 8 (c–d) highlight that the low accuracies of the model trained with generated data can be partially explained by the NN corruption.

## B.2 Avoiding DA$_{\text{IC-GAN}}$ on low quality classes

In this analysis, we try to exploit the observed correlation between per-class accuracy and per-class FID, i.e., samples quality (see Section 5.1.4), by restricting the use of DA$_{\text{IC-GAN}}$ only to classes that have an acceptable quality. We set an FID threshold of 150, under which we consider a class to have acceptable

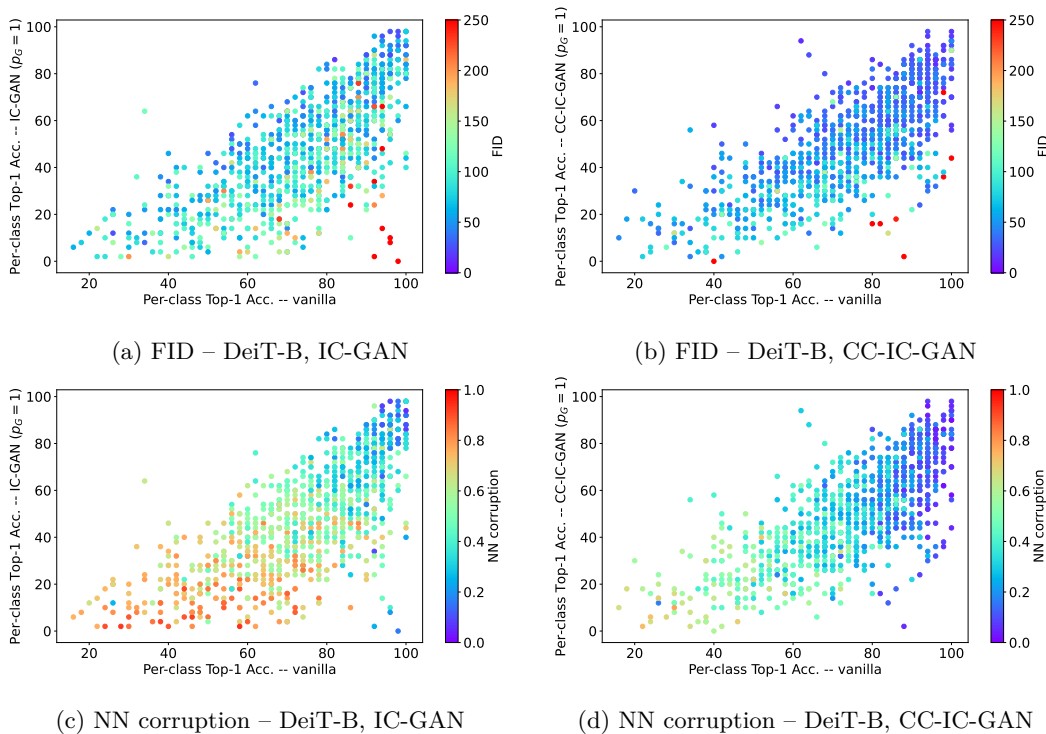

(a) FID – DeiT-B, IC-GAN

(b) FID – DeiT-B, CC-IC-GAN

(c) NN corruption – DeiT-B, IC-GAN

(d) NN corruption – DeiT-B, CC-IC-GAN

Figure 8: Impact of (CC-)IC-GAN's generation quality on per-class performance. (a-b) Per-class FID as a function of per-class top-1 accuracy of the vanilla and DA$_{\text{IC-GAN}}$ models. We observe that higher quality (CC-)IC-GAN generations tend to result in improved performances. (c-d) Per-class NN corruption as a function of per-class top-1 accuracy of the vanilla and DA$_{\text{IC-GAN}}$ models. We observe that less corrupted classes tend to result in improved performances. ImageNet validation results are shown for the DeiT-B model trained with horizontal flips, random crops, and RandAugment. We limited the FID colormap interval to 250 to aid interpretability, while we observed FID values up to 500 for certain classes.

quality as the visual inspection of classes with FID $>=$ 150 reveals either very poor image quality or mode-collapse (as shown in Figure 5); this threshold value is distant around $1.5\sigma$ and $3\sigma$ from the average per-class FID computed on IC-GAN and CC-IC-GAN samples, respectively. For this experiment, we train ResNet-152 with an augmentation recipe composed by HFlip and RRCrop applied to all classes, and DA$_{\text{IC-GAN}}$ applied to FID-filtered classes. We report the results in Table 6.

Table 6: ImageNet classification accuracy of ResNet-152 when using DA$_{\text{IC-GAN}}$ indistinctly on all classes vs. augmenting only classes with FID $<$ 150. For each column of results we report the mean top-1 accuracy computed over the indicated set of classes.

| Method | DA base | DA$_{\text{IC-GAN}}$ | Top-1 accuracy | | |
|---|---|---|---|---|---|
| | | | all classes | classes w/ FID $<$ 150 | classes w/ FID $>=$ 150 |
| ResNet-152 | HFlip + RRCrop | w/ IC-GAN | 77.71 | 77.44 | 80.44 |
| | | w/ FID-filtered IC-GAN | **77.94** | **77.56** | **81.67** |

Overall, we obtain, on average, a slightly better top-1 accuracy, +0.2%p, which can be stratified into +1.2%p considering the classes with FID $>=$ 150 and +0.1%p on the remaining classes. From these results we can observe that skipping the use of DA$_{\text{IC-GAN}}$ on poorly modeled classes increases the performances on such classes, while not harming the performance on the others.

## C    Additional Visualizations

Figure 9 displays images resulting from the combination of DA$_{\text{IC-GAN}}$ with the multi-crop augmentation used in the SwAV model Caron et al. (2020). As shown in the figure, these augmentations result in significant variations of the original image, with small crop images notably differing from the IC-GAN generations.

Figure 10 displays (CC-)IC-GAN generations. Note that IC-GAN and CC-IC-GAN generations (a–b) tend to show slightly different viewpoints and instances of the object present in the conditioning image (left-most column).

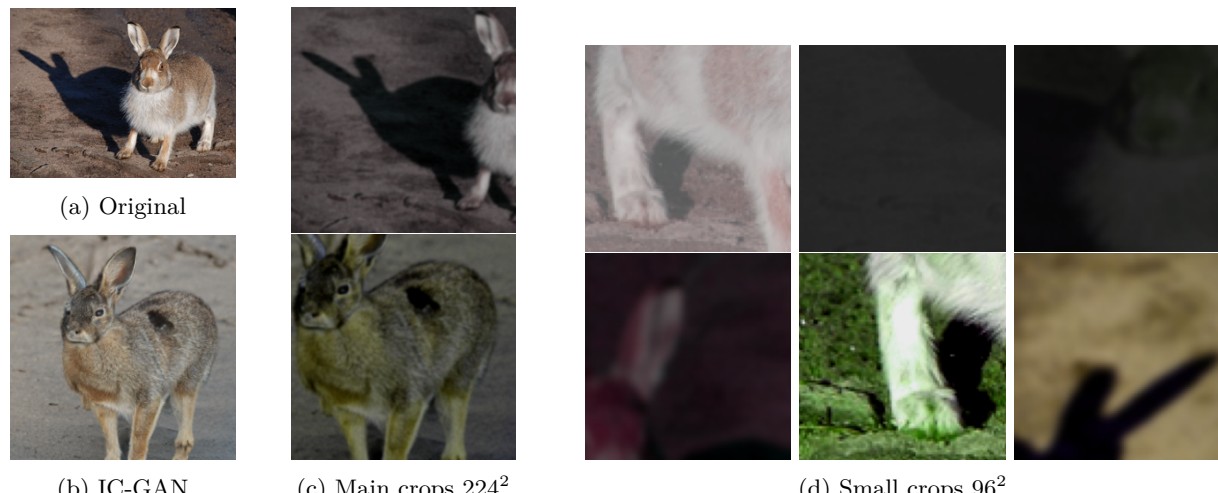

(a) Original

(b) IC-GAN          (c) Main crops $224^2$                              (d) Small crops $96^2$

Figure 9: Example of DA$_{\text{IC-GAN}}$ combined with multi-crop (Caron et al., 2020) augmentation with 2 main crops and 6 small crops: (a) depicts the original image, which is used to condition the IC-GAN generation process; (b) displays an IC-GAN generation; (c) shows the main crops of both images; and (d) presents the small crops obtained from the original image.

"warthog"

"black swan"

"tibetan terrier"

"tiger beetle"

"beer glass"

"cliff dwelling"

"hook'

"slot"

"water tower"

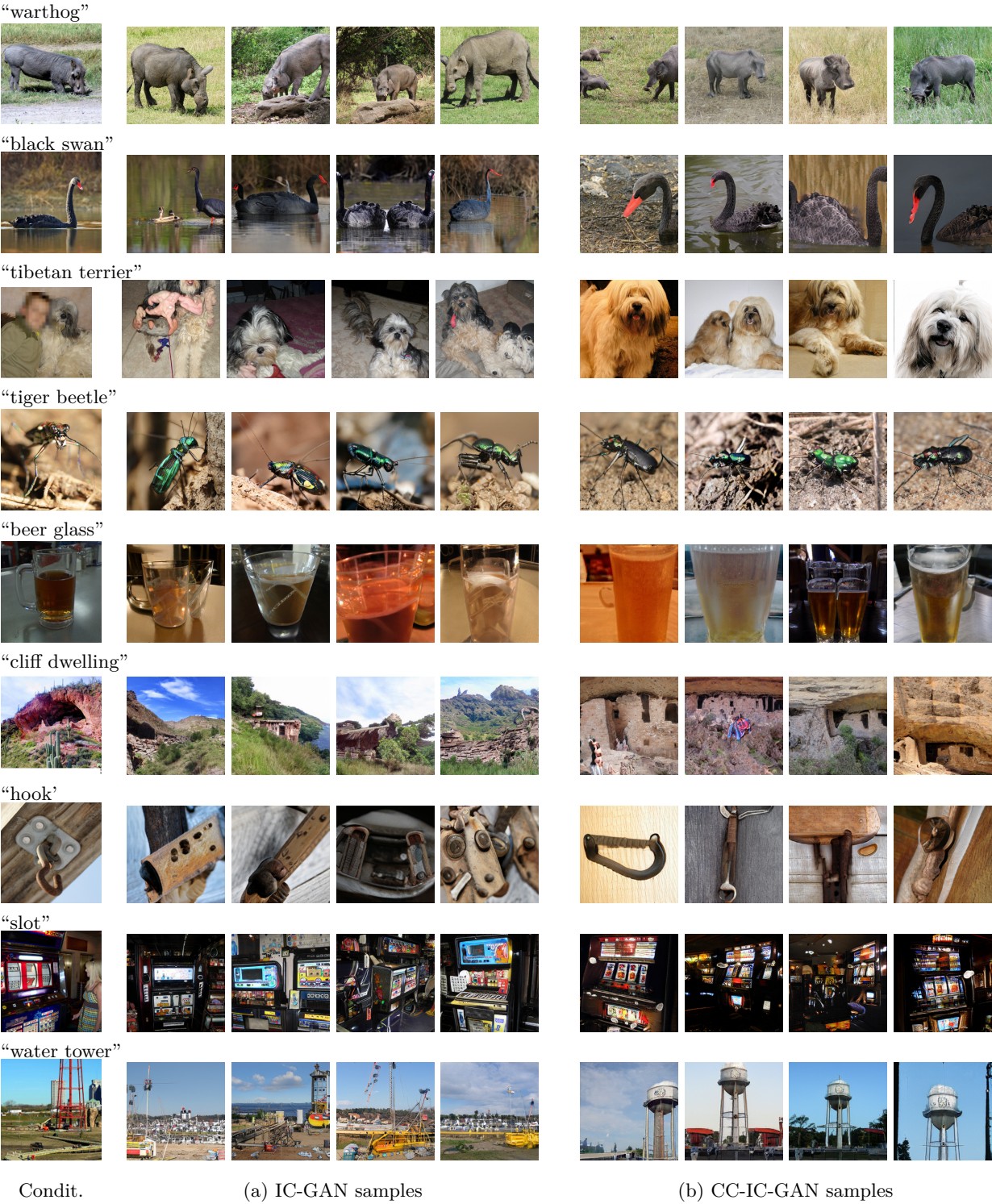

Condit.          (a) IC-GAN samples                    (b) CC-IC-GAN samples

Figure 10: Visual examples of (CC-)IC-GAN generations. Each row shows, from left to right, the conditioning image – i.e., central crop of ImageNet image –, followed by IC-GAN (a) and CC-IC-GAN (b) generated samples. Generetad samples were obtained using the depicted image conditioning and different noise vectors.

