# OpenReview forum: "Instance-Conditioned GAN Data Augmentation for Representation Learning"
_TMLR — Rejected by TMLR_

### Review · Reviewer_HrHE · 2022-08-17

**Summary Of Contributions:**

In this paper, the authors investigate leveraging the usage of Instance-Conditioned GAN (IC-GAN) in the context of data augmentation. In specific, a pre-trained generative model is deployed to generate the augmented samples conditioned on each input. The experimental results on ImageNet are given. The ConvNets and vision Transformers are considered in terms of both supervised and self-supervised learning.

**Broader Impact Concerns:**

I do not have concerns about this issue.

**Requested Changes:**

See the weaknesses above.

**Strengths And Weaknesses:**

Strengths:

- In general, I really like this paper. The research topic is critical. Once we can use GAN to generate decent training data, we are able to leverage almost an infinite number of training samples.

- The number of experimental results is impressive.

Weaknesses:

- However, unfortunately, I feel difficult to recommend the current version of the paper for acceptance. My main concern lies in empirical effectiveness. The technique proposed in the paper appears not to be effective enough. For example, as shown in Figure 3, it is incompatible with CutMixUp, one of the most widely-used regularization techniques. An analysis or improvement should be presented on top of this issue.

- Given that IC-GAN is proposed in previous works, and the limited empirical effectiveness, I feel that the contributions of this paper are insufficient.

- The results with unsupervised learning are interesting, but only a small number of experiments are conducted. Will the proposed method work on the recent masked autoencoders (MAE)?

- The results in Figure 5 are appealing, can we 'skip' the classes with poor image quality for data augmentation?

---

> ### Author Response · Authors · 2022-09-20
> **Answer to reviewer HrHE**
>
> We would like to thank the reviewer for the feedback. We are happy to read that the reviewer enjoyed reading the paper, finds the questions tackled in the paper to be of crucial importance for the Machine Learning community and is impressed by the number of the experiments conducted to support our claims. Below we answer the reviewer’s critiques.
>
> **Contribution and significance of CutMixUp results critiques (first two weaknesses mentioned by the reviewer)**: See the general comment.
>
> **Additional unsupervised learning experiments:** The IC-GAN transformation could be used with MAE models off-the-shelf. However, given the already large number of performed experiments and that the main focus of our analysis is the supervised learning scenario, which we complement with some SSL experiments, we decided to leave additional SSL experiments such as evaluating the effect of the IC-GAN based data augmentation on MAE for future works.
>
> **Excluding poor image quality classes:** We thank the reviewer for the suggestion. We ran additional experiments training ResNet-152 with HFlip, RRCrop, and DA_IC-GAN only applied on classes having per-class FID < 150 (909 classes out of 1000); The FID threshold of 150 is distant 1.5\sigma from the mean per-class FID computed over IC-GAN generations, and the visual inspection of classes with FID >= 150 shows either very poor image quality or mode-collapse (as shown in Figure 5 of the paper). The results are compared with the baseline model that applies DA_IC-GAN indistinctly to all classes, as shown in the following table.
> | Method                            | Top-1 acc.  (all classes) | Top-1 acc. (classes w/ FID < 150) | Top-1 acc. (classes w/ FID >= 150) |
> |-----------------------------------|---------------------------|-----------------------------------|------------------------------------|
> | RN-152 w/ DA-IC-GAN               | 77.71                     | 77.44                             | 80.44                              |
> | RN-152 w/ FID-filtered DA_IC-GAN  | **77.94**                     | **77.56**                             | **81.67**                              |
> | | | | |
>
> Overall, we obtain, on average, a slightly better top-1 accuracy, +0.2%p, which can be stratified into +1.2%p considering the classes with FID >= 150 and +0.1%p on the remaining classes. From these results we can observe that skipping the use of DA_IC-GAN on poorly modeled classes increases the performances on such classes, while not harming the performance on the others. We have included this table in Appendix B.2 of the paper. Also, the results of the same experiment but using CC-IC-GAN will be added as soon as they will be available.

---

### Review · Reviewer_mDGd · 2022-09-02

**Summary Of Contributions:**

The paper proposes to use Instance-Conditioned GAN (IC-GAN) to generate data augmentation (DA) samples around each image instance. Such samples are used for regularising the training of supervised image classifiers and self-supervised image encoders. The paper identifies cases where the IC-GAN helps or does not help the training.

**Requested Changes:**

Please address the weaknesses above.

Soundness
- Explain why and how the specific four invariances would be related to the specific set of robustness experiments reported.
- Either clarify the Fake-IN setup or remove the Fake-IN evaluations.
- Please fix the confusion between correlation and causation for sample quality and model accuracy.

Interestingness
- Clarify the conclusion of the paper.
- Explain why IC-GAN augmentation would still be interesting compared to CutMix and Mixup baselines.

Minor
- Use percentage points (%p or pp) instead of percentage (%) to express the percentage difference.

**Strengths And Weaknesses:**

## Strengths

The paper runs an extensive set of experiments to generate observations and recommendations for the readers. The paper does not try to oversell the method by falsely claiming its superiority over other baseline methods. The paper is generally organised well.

## Weaknesses - soundness

There are a few points in the experiments and the corresponding analyses with a lack of soundness in the arguments.

### Study of invariances against instance, viewpoint, occlusion, illumination.

The authors try to explain the apparent robustness of the IC-GAN trained models based on their invariance against a few factors. I don't think the paper makes a convincing case how and why such invariances to the four particular factors (instance, viewpoint, occlusion, illumination) have any relationship with the robustness measured in benchmarks like IN-A, IN-R, IN-ReaL, and ObjectNet.

### Fake-IN evaluations are not legitimate out-of-distribution evaluations.

When a classifier is trained on augmented samples from IC-GAN, it is not very meaningful to report results on the test samples on the "out-of-distribution" samples generated from IC-GAN (Fake-IN). They are not "out-of-distribution" for that classifier as they are included in the training set. Likewise for Fake-IN-CC.

### Confusing correlation with causation - augmented samples qualities and their impact on trained models.

Correlation does not imply causation, yet the authors imply this when they discuss the visual quality of generated samples and the generalisation performance of models trained on them. One should watch out for possible *common causes*. I can already suggest two hypotheses:

1. Visual *cleanliness* of a class. If a class is visually crisp, then the class will contain relatively less label noise and the visual attributes will be well-separated from the other classes. This will make both classification and generation easier for that class.
2. The number of samples per class. If there are many samples for a class, this will indeed make both classification and generation easier for that class.

## Weaknesses - interestingness

I'm not quite sure if the paper will be interesting to the TMLR audience.

### Why would anyone use IC-GAN instead of simple and strong augmentation methods like CutMix and Mixup?

Purely from the performance perspective, CutMixUp seems to beat the IC-GAN based augmentation variants. CutMix and Mixup results are also not fully reported on ResNet variants, making it hard to claim whether or not IC-GAN is still advantageous against those baselines.

Now from more practical point of view, I fail to find a reason why anyone would be interested in using IC-GAN rather than CutMix or MixUp. Using IC-GAN for augmentation is simply too complicated and expensive. There are many design parameters like augmentation probability, z-thresholding, hyperparameter tuning etc.

Also for ImageNet, it is fortunate that people have extensively studied the best training recipes and have open-sourced pre-trained GANs. They can really be downloaded immediately and be ready to use. If you'd like to train your model on a non-ImageNet dataset, the difficulty of training a GAN from scratch and searching for the right hyperparameters will increase drastically. It's much more practical and interesting to use simple (albeit hand-crafted) augmentation methods like MixUp and CutMix.

### What is the conclusion?

From the title, abstract, and intro, I thought the paper was recommending the researchers/practitioners use IC-GAN based augmentation for image classifiers and self-supervised image encoders. But the results are saying IC-GAN sometimes brings gains on top of horizontal flip and random resize crop, sometimes not. Compared to CutMix and MixUp, there is a lack of evidence that IC-GAN augmentation is in any way advantageous. Considering the complexity, computational costs, and adaptability to new datasets, this is even more so. I'm not quite sure if the paper is trying to advocate or discourage the use of IC-GAN for augmentation. It is good to be transparent about mixed results, but I belive a paper still needs to have a conclusion to be interesting to the audience. This is also reflected in the long list of 6 contributions on page 2 that are very difficult to grasp.

---

> ### Author Response · Authors · 2022-09-20
> **Answer to reviewer mDGd -- Part 1**
>
> We thank the reviewer for the detailed and punctual comments. We are pleased that the reviewer appreciated our empirical effort in running several experiments as well as our scientific transparency in presenting the results and providing suggestions or warnings for the reader. Here below, we reply to the weakness points reported by the reviewer.
>
> **Interestingness for the TMLR audience:**  We already commented on this point in the general response. However, we discuss the points raised by the reviewer:
>
> 1. **Unclear conclusions:** We agree with the reviewer that a paper must provide some clear conclusions. In our paper, we claim one main finding as in the last paragraph of the conclusions: “ To conclude, we have shown that current state-of-the-art large capacity models can be improved using instance conditioned generative models such as IC-GAN in conjunction with hand-crafted data augmentation techniques. ” But we have also warned the reader about the observed limitations, e.g., “in case of more aggressive data augmentation techniques for which DA_IC-GAN does not provide an improvement over the baselines, such as CutMixUp or multi-crop, we hypothesize that those augmentation recipes already result in strong image variations, and consequently, combining those with (CC-)IC-GAN generations out-of-the-box through DAIC-GAN may cause an over-regularization of the training.” Finally, given the reviewer's feedback, we decided to modify our list of contributions (end of Section 1) that now reads as follows: *“Overall, the contributions of this work can be summarized as follows:*
>    -  *We introduce DA_IC-GAN, a data augmentation module that combines IC-GAN with handcrafted data augmentation techniques and that can be plugged off-the-shelf into most supervised and self-supervised training procedures.*
>    - *We find that using DA_IC-GAN in the supervised training scenario is beneficial for high-capacity networks, e.g., ResNet-152, ResNet-50W2, and DeIT-B, boosting in-distribution performance and robustness to out-of-distribution when combined with traditional data augmentations like random crops and RandAugment.*
>    - *We extensively explore DA_IC-GAN’s impact on the learned representations, we discover an interesting correlation between per-class FID and classification accuracy, and report promising results in the self-supervised training of SwAV when not used in combination with multi-crop.*
>    - *We release the code-base and trained models at \url{anonymous.url} to foster further research on the usage of IC-GAN as a data augmentation technique.”*
> 2. **IC-GAN is more complex and limited to ImageNet domain compared to any strong hand-crafted augmentations like CutMixUp:** We identify two distinct critiques in this comment: (i) IC-GAN complexity and (ii) limitation to ImageNet. (i) We do agree with the reviewer about the perceived complexity of training a generative model like IC-GAN; however, there are many pre-trained models readily available that can be used out-of-the-box. Moreover, a trained generative model provides much richer and more plausible augmentations compared to handcrafted augmentations such as CutMix and MixUp – plus, it can be improved over time through re-training. We address the critique to the number of tunable hyper-parameters of DA_IC-GAN, by stating that it is not higher compared to handcrafted augmentations; in practice, we can limit the tuning to the single hyperparameter of “augmentation probability” while keeping the “truncation of z” safely chosen close to 1. Unfortunately, the need to tune other hyper-parameters already present in the training recipe is not mitigated by our augmentation module, but we show DA_IC-GAN having a positive impact in many settings where we did little to no hyper-parameters search. (ii) We do not present experiments where DA_IC-GAN is used to train a model on non-ImageNet datasets. However, according to the IC-GAN paper (Section 3.3 of Casanova et al., 2021), despite the pre-training is limited to ImageNet data, IC-GAN should be able to generate good quality images also when conditioned on out-of-distribution images, e.g., MS-COCO images.

---

> > ### Author Response · Authors · 2022-09-20
> > **Answer to reviewer mDGd -- Part 2**
> >
> > **Soundness:**
> >
> > - **Explain why and how the specific four invariances would be related to the specific set of robustness experiments reported:** The invariances reported in Section 5.1.3 and the robustness benchmarks from Section 5.1.2 should be treated as two independent experiments. Note that the datasets used for computing invariances are carefully picked (or designed) by Purushwalkam et al, (2020) and it is not possible to compute classification accuracy directly on them. Similarly, it is not trivial to compute the invariances on the robustness datasets  (IN-Real, IN-A, IN-R, ObjectNet, and Fake-IN). Thus, the experiments reported in Sections 5.1.2 and 5.1.3 are designed to probe the learned representations independently and should not be read jointly. If the reviewer could provide us with pointers to the sentences in the draft that they find misleading we would be happy to clarify them.
> > - **Clarify the Fake-IN setup:** We agree with the reviewer's observation that it might be misleading to refer to Fake-IN dataset as an out of distribution dataset w.r.t. the Imagenet dataset, since this dataset should model perfectly the ImageNet distribution having the IC-GAN model been successful during the learning. Note that since we use this dataset (and other datasets) to measure the robustness of the learned representations beyond imagenet dataset we decided to refer to these datasets as “robustness datasets” without directly implying if they are in distribution or out of distribution.
> > - **Confusion between correlation and causation for sample quality and model accuracy:** We agree with the reviewer, and we are aware that correlation does not imply causation. For this reason, in the paper we tried to never imply causation between sample quality and model accuracy, but we have always limited our claims to highlight the correlation between them. When putting the results into a broader context, we were careful to use vocabulary such as “potentially”, “we hypothesize”, “could”, etc, to avoid misinterpretations. However, if the reviewer could point us to the claims they believe could be misunderstood, we would be happy to clarify them.

---

### Review · Reviewer_Au9h · 2022-09-12

**Summary Of Contributions:**

The paper proposes to use Instance-Conditioned GANs [1] (IC-GAN) as data-augmentation technique to improve the performance of supervised and self-supervised models. The IC-GAN is added to the training pipeline and is combined with other augmentation methods such as horizontal flips and random crops to obtain augmented images. The approach is tested with ResNets and and DeiT-B trained on ImageNet in a supervised fashion and on a ResNet trained in a self-supervised fashion on ImageNet. Based on the evaluation on ImageNet accuracy the approach can lead to higher accuracy under specific circumstances. Additional evaluation is done to evaluate the robustness of learned representations and the effect of the IC-GAN's quality on the learned representations.

[1] Casanova, Arantxa, et al. "Instance-conditioned gan." NeurIPS, 2021

**Requested Changes:**

I am not sure there is much that can be done about the weaknesses. The evaluation is reasonably extensive but does not show clear improvements of using the approach compared to other, cheaper data augmentation pipelines.
Adding more novelty to the paper might fix this but I am not sure what kind of novelty we would be looking at, likely it would have to be a completely new approach.

**Strengths And Weaknesses:**

Strengths:
* the paper is well written and easy to understand
* the empiric evaluation is extensive and does not focus solely on accuracy improvements but also includes evaluation of the learned representations' robustness, as well as the effects of IC-GAN generation quality per class
* adding pretrained generative models to increse the performance of discriminative models is an interesting approach (which has been examined previously) to potentially benefit from large, pretrained models

Weaknesses:
* there is relatively little novelty; a pre-trained IC-GAN is added to exiting data augmentation techniques to essentially add one more data augmentation, but I can see no other novelty
* the overall results are mixed and I am not convinced the slight improvements in some settings justify the (relatively expensive) use of the IC-GAN during training
* most of the time, the overall best training performance is achieved with traditional data augmentation techniques (which don't rely on IC-GAN),
    * for the supervised DeiT-B using the full data augmentation pipeline usually leads to better performance than using IC-GAN
    * for the supervised ResNet IC-GAN does lead to some improvements in some settings but it is only compared against a relatively simple data augmentation pipeline (only horizontal flips and random resized crops)
    * for the semi-supervised setting the IC-GAN only leads to improvements if no multi-crop augmentation is used

Overall, I am not convinced that using the IC-GAN is actually worth the increased cost during training since many other data augmentation pipelines exist that achieve similar or better results without relying on external neural networks. Additionally, there is relatively little novelty since a pre-trained IC-GAN is simply added as a data augmentation building block to supervised training pipelines. The evaluation of the IC-GAN's generation quality on different classes is interesting but somewhat orthogonal to the goal of the paper.

---

> ### Author Response · Authors · 2022-09-20
> **Answer to reviewer Au9h**
>
> We thank the reviewer for the provided feedback. We appreciate the comments on the clarity of the paper, the extensiveness of our experimental results and that the reviewer found the approach interesting.
>
> We address the concerns regarding novelty and mixed results in the posted general comment. Here below, we reply to other critiques not yet covered:
>
> **Not convinced with the improvements of using DA_IC-GAN with respect to the additional computational cost:** We agree with the reviewer that DA_IC-GAN cannot be comparable to hand-crafted data augmentation when it comes to accuracy improvement and computational cost. However, the advantage of sampling augmentations from a generative model is about obtaining different images which are still realistic and semantically similar to the original [conditioning] image; this is in contrast to the hand-crafted data augmentations. This advantage could, in the future, lead to not relying anymore on hand-crafted augmentations, and from this perspective DA_IC-GAN provides a step towards it. In particular, our approach is an improvement in terms of inference cost compared to previous methods based on GAN-inversion and for impact on larger-scale datasets (ImageNet) not yet investigated by the community. We believe this positions our work as an interesting read to the TMLR community and our exploration can be helpful to speed up further research on this topic.
>
> **Evaluation of IC-GAN’s generation quality is orthogonal to the paper:** We believe that analyzing the image quality of IC-GAN classes is relevant to our proposed approach as it allows us to conjecture a relation between performance and sample quality, and eventually extend the analysis by skipping DA_IC-GAN augmentation on low-quality classes. We report this additional result in the comment to reviewer HrHE as well as Appendix B.2 of the paper.

---

### Author Response · Authors · 2022-09-20
**General response to the reviewers**

We thank all three reviewers for the time spent on our manuscript and for their feedback. We first address a common critique by all three reviewers about the significance of our results and the novelty aspects of the manuscript. We address the remaining comments in the individual answers to each reviewer.

**General comment about the manuscript significance and novelty.**

All three reviewers are concerned about the significance of the results as the manuscript presents mixed experimental results – as nicely summarized by reviewer **Au9h** – as well as limited novelty – as the paper introduces a data augmentation that is using a pretrained IC-GAN model that has already been published in a different venue –.

To address these critiques, we would like to bring to the reviewers’ attention the acceptance criteria for TMLR journal that can be found under [this link](https://www.jmlr.org/tmlr/editorial-policies.html) (see evaluation criteria):
> The acceptance decision for a submission is based on the answers to the following questions:
> - Are the claims made in the submission supported by accurate, convincing and clear evidence?
> - Would at least some individuals in TMLR's audience be interested in knowing the findings of this paper?
>
> Papers should be accepted if they meet the criteria, even if the contribution or significance of the work is modest.

Thus, when deciding about paper acceptance for TMLR journal one should not judge the significance (of the results) and the novelty aspects of the method presented in the manuscript. That said, based on the TLMR acceptance criteria, one should judge (1) the quality of the empirical support for the claims made in the paper and (2) the interest for the TMLR community (even if the findings are mixed).

**Re point #1**: Only reviewer **mDGd** has some minor doubts about the correctness of some parts of our manuscript and we address the reviewer’s doubts in the individual answer. Nevertheless, the same reviewer highlights the correctness of our claims: “The paper does not try to oversell the method by falsely claiming its superiority over other baseline methods.” The remaining reviewers do not criticize the manuscript from the point of view of the claims made in the paper. Moreover, all reviewers appreciate the breadth of our analysis: [**HrHE**] “The number of experimental results is impressive”; [**mDGd**] “The paper runs an extensive set of experiments to generate observations and recommendations for the readers”; [**Au9h**] “the empiric evaluation is extensive and does not focus solely on accuracy improvements but also includes evaluation of the learned representations' robustness, as well as the effects of IC-GAN generation quality per class”.

**Re point #2**: All reviewers seem to appreciate the research question tackled in our manuscript: [**Au9h**] “adding pretrained generative models to increase the performance of discriminative models is an interesting approach…”; [**HrHE**] “In general, I really like this paper. The research topic is critical. Once we can use GAN to generate decent training data, we are able to leverage almost an infinite number of training samples.”. Reviewer **mDGd** has some doubts about the interest to the TMLR community but their hesitance seems to be solely based on the significance of the reported results. In regard to this, we first recall that one rule of thumb according to the TMLR board (taken from [this video](https://www.youtube.com/watch?v=Uc1r1LfJtds) min 10:50), is “...if reviewers are unsure as to whether a submission satisfies the [acceptance] criteria they should just assume that it does...”; secondly, but not for importance, we argue that our work tackles an important research question (use of generative model for representation learning) and provides an extensive analysis that “generates observations and recommendations for the readers” [**mDGd**]. Thus, although the overall results are mixed, we strongly believe our work is of interest for ML practitioners as well as for researchers exploring alternatives to handcrafted data augmentations and as such should be a great resource for “at least some individuals in TMLR's audience”. We address the concerns of reviewer **mDGd** in the individual answer.

Finally, the TMLR journal acceptance criteria provides examples of papers that should not be accepted, under the same link as above we can read:
> “Papers that should not be accepted include papers that make bold statements unsupported by empirical or rigorous evidence, papers that aren’t clearly written, papers that incorrectly claim novelty over existing published work, and papers that merely re-implement an idea that has already been reproduced before.”.

Based on the three submitted reviews, we believe that our paper does not fall under any of the categories mentioned above.

---

### Decision · Action_Editors · 2022-10-19

**Recommendation:** Reject

**Comment:**

The submission proposes a data augmentation module called DA_IC-GAN which uses the Instance-Conditioned GAN (IC-GAN) generative model to produce augmented examples from existing ones. Results are presented on ImageNet for supervised (ResNet, DeiT-B) and self-supervised (SwAV) models. Robustness, feature invariance, per-class, and ablation analyses are also provided. The paper concludes that for higher capacity networks and when coupled with hand-crafted augmentations, data augmentations obtained from instance-conditioned generative models such as IC-GAN improve classification performance. On the other hand, the paper shows the benefits of DC_IC-GAN disappear when coupled with more aggressive data augmentation techniques like CutMixUp or multi-crop.

Reviewers like the submission's writing quality (Au9h, mDGd), clarity (Au9h), and extensive empirical evaluation (Au9h, mDGd, HrHE). Most reviewers express interest in using pre-trained generative models to improve classification quality (Au9h, HrHE).

The bulk of reviewers' concern revolves around novelty (Au9h, HrHE) and general interest to the TMLR community (Au9h, mDGd, HrHE). In their response, the authors refer to TMLR's acceptance criteria, namely that a submission should make claims that are supported by accurate, convincing, and clear evidence, and that its findings should be of interest to at least some individuals in the TMLR audience. They (rightfully) argue that the submission's acceptance should not be decided on significance or novelty. My recommendation will therefore set aside those reviewer concerns and focus on clarity, correctness, and interest to the TMLR community.

All reviewers point out in one way or another that strong hand-crafted augmentations like CutMixUp and multi-crop remain superior to DA_IC-GAN in terms of model performance and computation. While on the surface this looks like a significance objection, I think the reviewers are asking "if not for performance improvements, what makes this submission interesting to the TMLR community?", which indicates that its conclusions and main claims are not clearly articulated from the reviewers' perspective. This is evidenced by reviewer mDGd asking "what is the conclusion?" Reviewer mDGd also questions the scope of DA_IC-GAN's appeal given that obtaining pre-trained generative models becomes more complicated as we move away from ImageNet.

In their response, the authors clarify the submission's main claim (with large capacity networks and combined with simpler hand-crafted augmentations, DA_IC-GAN improves model performance) and update the submission's list of claimed contributions. The authors also list advantages like producing realistic augmentations that are similar to the original image and removing the need for hand-crafted augmentations. Reviewer mDGd objects that they don't think it essential to generate richer and more plausible augmentations, and what ultimately matters is whether the augmentations result in better generalization. On the applicability of DA_IC-GAN beyond ImageNet, the authors refer to Casanova et al. (2021) to support IC-GAN's ability to generate augmentations conditioned on OOD images, like MS-COCO images. Reviewer mDGd questions how far out-of-distribution we can trust the empirical evidence.

Finally, reviewer mDGd expressed a few concerns on technical correctness. They remain skeptical of Fake-IN's characterization: they point out that even without explicitly referring to in-distribution or out-of-distribution, the term "robustness" itself implies some resilience against certain changes, and they ask against what the model is resilient if not out-of-distribution examples. They also object to the submission presenting a correlation between sample quality and model accuracy without discussing potential explanatory factors, wondering what should the readers learn from the shown correlation and what actions the readers can take based on those observations.

In light of the above, reviewers remain unconvinced that the submission's contributions are of sufficient interest to the TMLR community. I therefore recommend rejection.

**Audience:**

All reviewers point out in one way or another that strong hand-crafted augmentations like CutMixUp and multi-crop remain superior to DA_IC-GAN in terms of model performance and computation. While on the surface this looks like a significance objection, I think the reviewers are asking "if not for performance improvements, what makes this submission interesting to the TMLR community?", which indicates that its conclusions and main claims are not clearly articulated from the reviewers' perspective.

**Claims And Evidence:**

Reviewers object to the submission presenting a correlation between sample quality and model accuracy without discussing potential explanatory factors, wondering what should the readers learn from the shown correlation and what actions the readers can take based on those observations.